# Historical droughts manifest an abrupt shift to a wetter Tibetan Plateau

Yongwei Liu[1], Yuanbo Liu[1*], Wen Wang[2], Han Zhou[3], Lide Tian[4]

[1] Key Laboratory of Watershed Geography Sciences, Nanjing Institute of Geography and Limnology, Chinese Academy of Sciences, Nanjing, China

[2] State Key Laboratory of Hydrology-Water Resources and Hydraulic Engineering, Hohai University, Nanjing, China

[3] Wuhan University of Technology, Wuhan, China,

[4] Yunnan University, China

*Correspondence to*: Yuanbo Liu (ybliu@niglas.ac.cn)

**Abstract.** The Tibetan Plateau (TP) plays a vital role in Asian and even global atmospheric circulation, through the interactions
between land and atmosphere. It experienced significant climate warming and spatially and temporally variant wetting over the past half century. Because of the importance of land surface status to the interactions, determining the wetting/drying of the TP from individual changes in precipitation (Prep) or temperature is difficult. Soil moisture (SM) is the water synthesis of the surface status. The persistent deficit of SM (SM drought) is more sensitive to climate change than normal SM. This study first explored the climate wetting/drying of the TP from variations in historical SM droughts over 1961-2014, with a focus on
spatiotemporal patterns, long-term variations, and climate causes of summer (May–September) SM droughts based on multiple observation and reanalysis data. The results showed comparatively frequent and severe droughts in the central and southern area, particularly in the semi-arid and sub-humid regions. SM drought exhibited an abrupt and significant ($p<0.05$) alleviation in the interior and central-west TP in the mid to late 1990s. The prominent drought alleviation indicated a hydro-climate shift to a wetter plateau, not merely steady trends in the literature. We demonstrated that the wetting shift was dominated by Prep
over potential evapotranspiration (PET). By contrast, the in-phase trends were combing forces of Prep and PET, with increased forces of PET after the wetting shift. Furthermore, the Prep dominance was largely attributed to a phase transition of the Atlantic multi-decadal oscillation from cold to warm since the mid-1990s. The PET impacts on the wetting trends were likely dominated by solar radiation, wind speed, and vapour pressure deficit. Regionally, the wetting shift was distinct from the arid to semi-arid and semi-arid to sub-humid climate. Such spatiotemporal changes may affect the TP's atmospheric circulation
and, subsequently, the Asian monsoon and global circulation, in addition to the fragile ecosystem in the TP.

# 1 Introduction

The Tibetan Plateau (TP), with an area of ~2.5 million square kilometers and more than half at the altitude above 4000m in central Asia, is the highest and most extensive plateau in the world (Fig. 1). It therefore has long been described as "the roof of the world" and referred to by the scientific community as "the third pole". The TP and its surroundings contain the largest number of glaciers outside the polar regions and the largest concentration of inland lakes worldwide (Wan et al., 2016). It is also regarded as the "water tower of Asia", which is the birthplace of dozens of major rivers in Asia (e.g., Yangtze River, Huang River, Yarlung Zangbo River, Indus River, and Ganges). The TP exerts strong thermal forcing on the atmosphere (i.e., land-atmosphere feedback) over the Asian monsoon region (Yang et al., 2014). The climate change of the TP has profound impacts on the Asian and even global climate and hydrology.

The TP has undergone significant warming over the past decades (Liu and Chen, 2000; Yao et al., 2012). The air temperature (Temp) has increased by approximately 0.35 ℃ per decade since 1970 (Yao et al., 2019), at a warming rate of ~1.5 times the global surface air temperature increase (Zhang et al., 2013; Wang et al., 2008). The magnitude of climate warming increases from south to north (Yang et al., 2011; You et al., 2016). As a consequence of the greenhouse effect along with the increased absorbed energy via the snow-albedo feedback and cloud-radiation interactions, the TP's warming is accelerated (Liu and Chen, 2000; Duan and Xiao, 2015). The increasing Temp substantially affects the plateau's hydrological cycle, and the melting of glaciers, degradation of permafrost, increasing of river discharge, and expansion of lakes are among the most prominent effects of climate warming (Cheng and Wu, 2007; Yao et al., 2012; Cuo et al., 2014; Teng et al., 2021). The increasing Temp also increases the evaporative demand and consequently tends to contribute to the climate drying or aridity of the TP (Kuang and Jiao, 2016; Yao et al., 2019). Unlike the pronounced increase in Temp, the precipitation (Prep) did not show consistent, plateau-wide changes over the past decades (Yang et al., 2011). Although the major rain gauges and station measurements exhibited general increasing trends in Prep, the trends were variable in space and time with large heterogeneity (Yao et al., 2012; Kuang and Jiao, 2016). The Prep in the subregions is increasing (e.g., the west and central TP) in some and decreasing (e.g., the southeast TP) in others, accompanied by non-uniform changes in different seasons (Xie et al., 2010; Yang et al., 2011; Gao et al., 2014). Furthermore, the actual evapotranspiration (ET) generally follows the Prep changes under significant warming (Gao et al., 2007; Yin et al., 2013). Thus, the increase/decrease in Prep cannot effectively reflect the climate wetting/drying of the plateau. Under the significant overall warming up and variant Prep increases/decreases in space and time, the wetting/drying changes in the TP become complicated and remain unclear.

Although Prep and Temp are the focus of climate change, they cannot directly and effectively reflect the climate wetting/drying of the TP. The TP is a region with the strongest land-atmosphere coupling in the mid-lattitudes (Xue et al., 2017), where both Prep and Temp are insufficient to account for the impacts of the feedback from the underlying land surface to the atmosphere. Meanwhile, soil moisture (SM) is a pivotal link between the land surface and atmosphere (Wanders et al., 2014). It is also the largest storage component of Prep and radiation anomalies on land (Seneviratne et al., 2010). SM not only determines the allocation ratio of rainfall in surface flow and infiltration, but also couples with evaporation and controls the

water and energy feedback from the land surface to atmosphere. Specifically, SM modulates temperature variability through evaporation cooling (i.e., SM-Temp coupling) and influences Prep through the impacts on evaporation and boundary layer thickness (i.e., SM-Prep coupling) (Koster et al., 2011). Therefore, SM changes (drying/wetting) not only affect the regional underlying incoming water and energy redistribution but also produce impacts on large scale even global climate system (Koster et al., 2016). Moreover, SM anomalies can persist for months, with substantial impacts on water and energy fluxes on land. These properties make SM capable of comprehensively reflect climate anomalies and land surface feedback (Diro and Sushama, 2017). Compared with the abnormal high or normal SM status, the abnormal low SM status (i.e., SM deficiency/drought) is more sensitive to climate change, because the soil under low moisture status is more sensitive to Prep variation and has more control on evapotranspiration (Seneviratne et al., 2010). Hence, the SM deficiency (drought) with its variations could better reflect the climate wetting/drying of the TP under the coupled land and atmosphere (Lansu et al., 2020; Liu et al., 2021). However, according to our review of the literature, the climate wetting/drying of the TP has never been investigated from the perspective of variations in SM droughts. Little is known about the SM droughts of the TP, including not only the drought spatiotemporal patterns and long-term variations but also the climatic causes behind them.

Therefore, this study investigated the climate wetting/drying of the TP from the perspective of changes in historical SM droughts over the last half century. A synthetic investigation of SM droughts was conducted to (i) understand the spatiotemporal patterns and long-term variations in SM droughts on the TP and (ii) explore the climate causes behind them. This study assessed the SM droughts in summer (May–September). This practice is to ensure the reliability of SM drought identification, because SM is frozen or freeze-thawed during cold months (i.e., October – next April) on the TP. Large-scale accurate, reliable SM estimation cannot be achieved with limited knowledge of the freeze-thaw processes for both remote sensing retrieval and land surface modelling (Taylor et al., 2012; Zwieback et al., 2015; Liu et al., 2019a).

## 2 Study area and data

### 2.1 Study area

TP is the largest plateau in China and the highest in the world (Fig. 1a). The plateau is dominated by the Indian monsoon and the westerlies, with limited influence from the East Asian monsoon (Yao et al., 2012; Yang et al., 2014). The climate presents a general characteristic of low air temperature (Temp), strong solar radiation, and large spatial and temporal variability of Prep (Xie and Zhu, 2013; Xu et al., 2008). Limited by low Temp, the vegetation over most TP generally grows in summer (May–September) when the monthly average Temp is above 0℃ (Fig. S1). Spatial patterns of the monthly average Prep, Temp, and SM for May–September are shown in Fig. 1b-d.

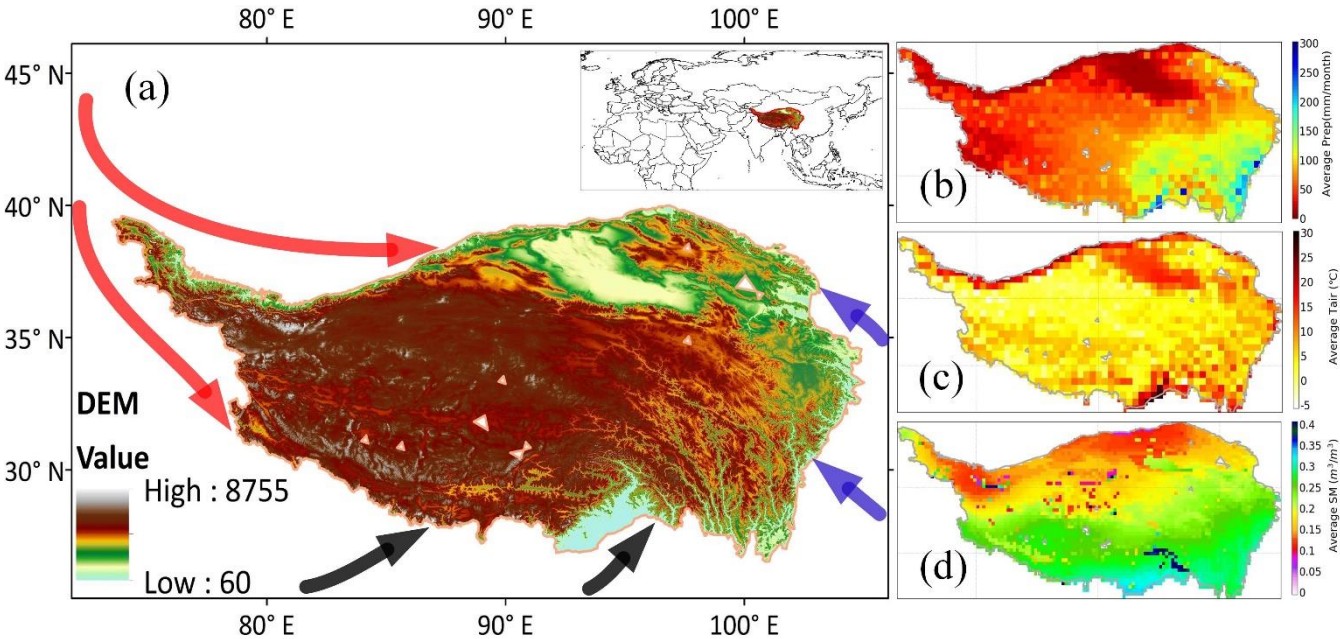

**Figure 1.** (a) Location of the Tibetan (TP) and the elevation; (b) Monthly average precipitation (Prep), (c) air temperature (Tair/Temp) and (d) soil moisture (SM) over the summer periods (May–September). Prep and Tair/Temp are based on the gauging interpolation data provided by the Chinese Meteorological Administration (CMA). SM is from the Noah model driven by the Global Land Data Assimilation System (GLDASv2.0/Noah) dataset with a depth of 0-10cm. The black, red, and blue arrows represent the Indian monsoon, the westerlies and the East Asian monsoon, respectively.

**2.2 Data**

To improve the reliability of the investigation, both the SM data generated by the Noah model driven by the Global Land Data Assimilation System (GLDASv2.0/Noah) and the fifth generation of the land component of the European Centre for Medium-Range Weather Forecasts atmospheric reanalysis (ERA5-Land) were used in this study. This decision is based on a large number of literature assessing the data quality of the existing SM products and the inconsistency in their results up to now, largely caused by such factors as the time period, location of in situ SM measurements, and data quality indicators in SM data evaluation (e.g., Su et al.0, 2011; Chen et al., 2013b; Zeng et al., 2015; Bi et al., 2016; Zhang et al., 2018, 2019a; Xing et al., 2021; Su et al., 2021; Zeng et al., 2022). For example, the SM data of the TP from GLDAS Noah was verified to be better than the ESA CCI SM from the European Space Agency's Climate Change Initiative project and the global atmospheric reanalysis product of ERA-Interim from the European Centre for Medium Range Weather Forecasts in Zhang et al. (2019a), but the ESA CCI SM was evaluated to be better than GLDAS Noah SM in Zeng et al. (2022). Moreover, the in situ SM observations from the International Soil Moisture Network (ISMN) were used to verify the validity of the SM data from GLDASv2.0/Noah and ERA5-Land. A total of 111 ISMN soil moisture stations are available from three observation networks

of NGARI, CTP-SMITN, and MAQU, representing arid, semi-arid and sub-humid, and humid climate, respectively (Figure S2). GLDAS Noah and ERA5-Land simulates SM at four depths with spatial and temporal resolutions of 0.25°, 3 h and 0.1°, 1h. In this study, the monthly resampled surface SM of 0.25° for GLDAS Noah and ERA5-Land SM over 1961-2014 was utilized.

Multi-elemental data of precipitation (Prep), air temperature (Temp), potential evapotranspiration (PET), surface (10 m) wind speed, downward shortwave radiation flux (Radi), vapor pressure deficit (VPD), latent heat flux, and net radiation flux were used to support the SM drought study. Prep and Temp were mainly provided by the Chinese Meteorological Administration (CMA) with a 1-month and 0.5° resolution. They were produced based on over 2400 national ground-based meteorological gauging stations using spatial interpolation approaches with elevation effects considered. PET, wind speed, Radi, VPD, latent heat flux, and net radiation flux were obtained from the output and forcing datasets of GLDAS Noah and ERA5-Land, respectively (Munoz-Sabater et al., 2018; Beaudoing et al., 2019). PET, wind speed, Radi, latent heat flux, and net radiation flux were directly extracted. VPD was calculated based on the specific humidity, surface pressure, and surface (2 m) air temperature from GLDAS forcing and ERA5 reanalysis datasets. GLDASv2.0/Noah was entirely forced by the Princeton meteorological forcing dataset (Sheffield et al., 2006). This dataset was constructed by combining a suite of global observation-based datasets with the National Centers for Environmental Prediction-National Center for Atmospheric Research (NCEP-NCAR) reanalysis. Specifically, the contributing datasets used in the development of the forcing dataset comprised the NCEP-NCAR reanalysis, Climate Research Unit, Global Precipitation Climatology Project, Tropical Rainfall Measuring Mission, and the National Aeronautics and Space Administration Langley Surface Radiation Budget Project. ERA5-Land is a replay of the land component of the ERA5 climate reanalysis, forced by meteorological fields from ERA5. In addition, the Prep from China Meteorological Forcing Dataset (CMFD) (He et al., 2020) and the ET developed by Han et al. (2020, 2021) (hereafter known as Han ET) were incorporated in the inter comparisons with that of GLDAS and ERA5 to support the validity of SM evaluation.

To explore the climate causes of SM drought variations, the circulation indices of the Atlantic multi-decadal oscillation (AMO), the Pacific Decadal Oscillation (PDO), and the El Niño-Southern Oscillation (ENSO) (Niño 3.4) from the National Oceanic and Atmosphere Administration (NOAA)–Physical Sciences Laboratory and the NOAA-National Centres for Environmental Information (NCEI) were used. The monsoon circulation index of the Indian summer monsoon (ISM) (Wang et al., 2001) and the East Asian summer monsoon (EASM) (Li et al., 2002) were adopted. The solar cycle was also accounted using the solar radiation flux indices (SRF). Additionally, the ERA5 geopotential and U-/V- components of wind data were used in the upper air analysis. Considering the fact that changes in vegetation coverage is closely related to the changes in soil moisture (Zhong et al., 20110), the Normalized Difference Vegetation Index-3rd generation (NDVI) from the Advanced Very High Resolution Radiometer (AVHRR) sensors under the Global Inventory Monitoring and Modeling System (GIMMS) was incorporated in the analysis.

## 3 Methodology

Drought is a sustained period of below-normal water availability (Tallaksen et al., 2004). It has a temporally slow accumulation and spatially continuous expansion process. With the deepening of drought understanding, drought identification has expanded from a single time dimension to space-time synchronous recognition (Andreadis et al., 2005; Lloyd-Hughes, 2012; Xu et al., 2015; Herrera-Estrada et al., 2017; Zhu et al., 2019). Space-time joint identification could consider the space-time continuous property of actual droughts and reconstruct the drought development process. Therefore, it could improve the reflection of the real characteristics of droughts (e.g., the spatial position/overall severity/duration of drought) and benefit the investigation of the climate forces of droughts (Lloyd-Hughes, 2012; Liu et al., 2021). In this study, the space-time joint identification of droughts was realized by recognizing the drought clusters (the spatially continuous regions under drought for each month) and eventually the drought events (the spatially continuous and temporally overlapped domain under drought).

### 3.1 Drought event identification

The space-time joint identification of drought events comprises the following steps:

(1) Identification of the drought conditions for each SM grid pixel. The percentile was used as the drought indicator because of its effectiveness in many large-scale drought assessments (e.g., Herrera-Estrada et al., 2017). The monthly SM percentile on each grid was calculated from the empirical cumulative distribution function of each month (May–September) separately, based on the monthly SM from 1961 to 2014. Then, the monthly map of the SM percentile was smoothed using a 2-dimensional median filter to reduce the spatial noise. Grids with an SM percentile less than 0.15 are regarded as being under drought, because it approximates the common drought metrics of the standard precipitation index value of -1.0 (Guttman, 1999).

(2) Determination of the spatial continuity of grid pixels under drought. The spatially continuous grids below the drought threshold belong to one drought cluster. Drought clusters were identified using a spatial clustering method (Andreadis et al., 2005). Small clusters of less than 1.5% of the TP area were not accounted for in case of spurious long droughts caused by tenuous spatial continuity (Liu et al., 2019b; Liu et al., 2021).

(3) Recognition of the temporal continuity of drought clusters. The temporal links of drought clusters between adjacent months were determined according to their overlapping areas. Drought clusters with overlapping areas over 625 km$^2$ (i.e., one grid pixel) were classified as being in the same drought event (Herrera-Estrada et al., 2017; Liu et al., 2021).

### 3.2 Drought characterization

The following drought characteristics are accounted for in the space-time joint identification of drought events in a 3-dimensional (latitude-longitude-time, 3-D) domain.

(1) Drought duration: the time interval between the onset and the ending of one drought event.

(2) Drought area: the total area affected by a drought in its duration.

(3) Drought intensity and severity reflect the strength of drought and the water deficiency of one drought event, respectively. Drought intensity is represented by the value of 1.0-percentile for each grid pixel and characterized by the area average intensity of all grids within the drought cluster for each drought cluster (Herrera-Estrada et al., 2017). Drought severity can be expressed as the cumulative value of drought intensity for all grids within the space-time continuous structure of one drought event.

(4) Drought cluster/event centroid: being defined as the weighted centre of the grid intensity, with the intensity value at each grid within the drought cluster/event domain as weights. For more details of the aforementioned drought identification and characterization, please refer to our previous research (Liu et al., 2021).

## 4 Results

### 4.1 Validation of soil moisture data from GLDAS Noah and ERA5-Land

The validity of GLDAS and ERA5 SM was verified based on the in situ SM observations from the ISMN. The 111 ISMN stations with a data record of 2008-2014 were divided into $0.25° \times 0.25°$ grids. Consequently, 26 measured grids in total were identified, with 5 in the arid NAGARI, 12 in the semi-arid and sub-humid CTP-SMITN, and 9 in the humid MAQU. The mean SM value of each grid was obtained by averaging the measurements of all stations falling within that grid pixel. The GLDAS and ERA5 monthly SM were compared with the ISMN measurements over summer periods of May-September from 2008 to 2014 on 25 grids (1 grid without measured data in NAGARI).

Generally, GLDAS shows lower bias and root mean square error (rmse), but higher unbiased rmse (ubrmse) and lower Pearson correlation coefficient (r) than ERA5 (Fig. 2). ERA5 SM seems more consistent with in situ observations in temporal variations than GLDAS SM during summer periods. Specifically, Fig. 3 shows that: for MAQU located in humid region, GLDAS SM performs better in bias, but ERA5 is better in r and ubrmse; For NGARI in arid region, ERA5 is better in bias, r, and rmse, but worse in ubrmse than GLDAS; For CTP-SMTMN in semi-arid and sub-humid region, GLDAS shows larger advantage in bias and rmse, but ERA5 has more advantage in r and ubrmse. ERA5 SM seems more advantageous than GLDAS in semi-arid, sub-humid, and humid region without the system bias considered. Both ERA5 and GLDAS SM presents considerably higher temporal consistency with in situ measurements in the arid, semi-arid, and sub-humid than humid region (Fig. 3). It may indicate more credibility in SM variation analysis for the arid, semi-arid, and sub-humid regions.

The validity of GLDAS and ERA5 SM was also assessed from the inter comparisons of Prep and ET considering that SM variations are largely controlled by the rainfall and evapotranspiration process (Su et al.,2018). The inter comparison focuses on temporal consistency of different datasets based on Pearson's correlation analysis. The results show that both GLDAS and ERA5 Prep present higher temporal consistency with CMA and CMFD Prep in south and east TP than north and west TP (Fig. 4). Generally, ERA5 Prep has better temporal consistency with CMA and CMFD prep than GLDAS, particularly in southwest TP (Fig. 4 a,c,d,f). In addition, the GLDAS and ERA5 ET has high temporal consistency in most TP. The inconsistency mainly

in the northwest and south fringe of the plateau (Fig.5a). The ERA5 ET agrees better with Han ET than that of GLDAS, in particular for the southwest regions (Fig. 5b,c). The ET in the northwest, north, and southeast seems bearing larger uncertainty than other regions. The aforementioned performance of Prep and ET for GLDAS and ERA5 indicates the possibility of low credibility in SM modelling in parts of northwest, north and southeast TP.

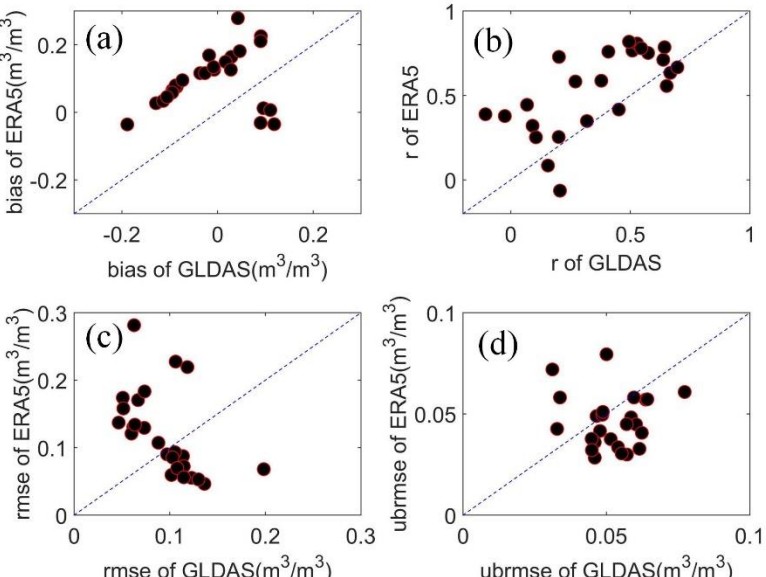

**Figure 2.** Comparisons of the (a) bias, (b) Pearson correlation coefficient (r), (c) root mean square error (rmse) and (d) unbiased rmse (ubrmse) for GLDAS and ERA5 soil moisture in summer periods of May-September over 2008-2014 based on the soil moisture measurements from the ISMN.

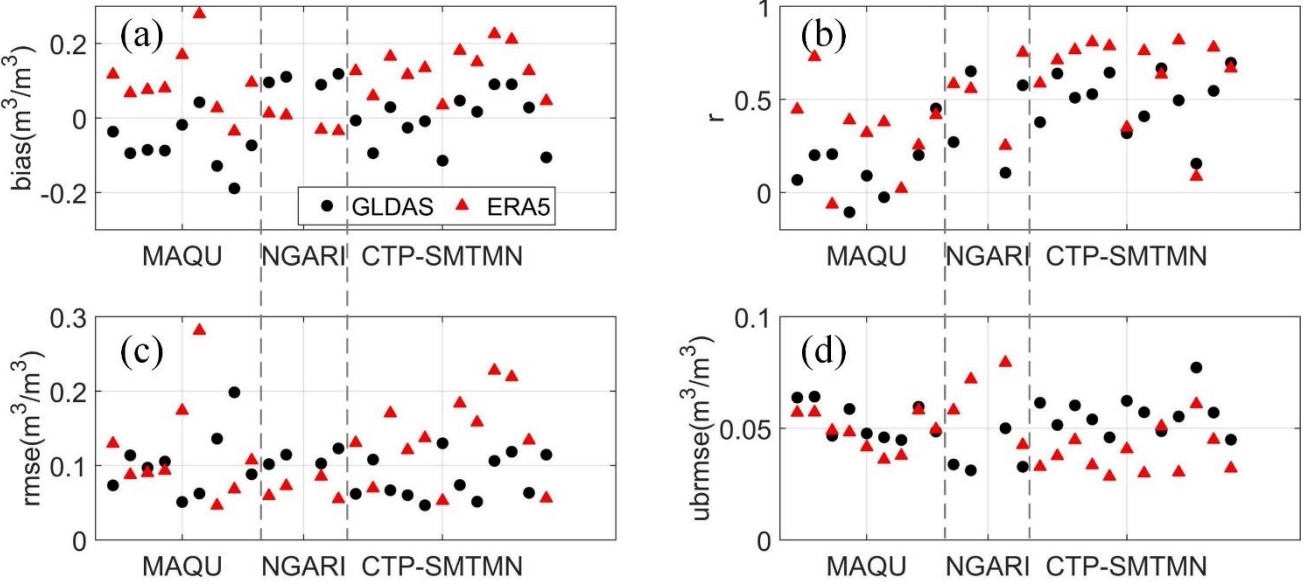

**Figure 3.** The (a) bias, (b) Pearson correlation coefficient (r), (c) root mean square error (rmse) and (d) unbiased rmse (ubrmse) for GLDAS and ERA5 soil moisture in summer periods of May-September over 2008-2014 based on the ISMN station measurements from the MAQU, NGARI, and CTP-SMTMN observation network.

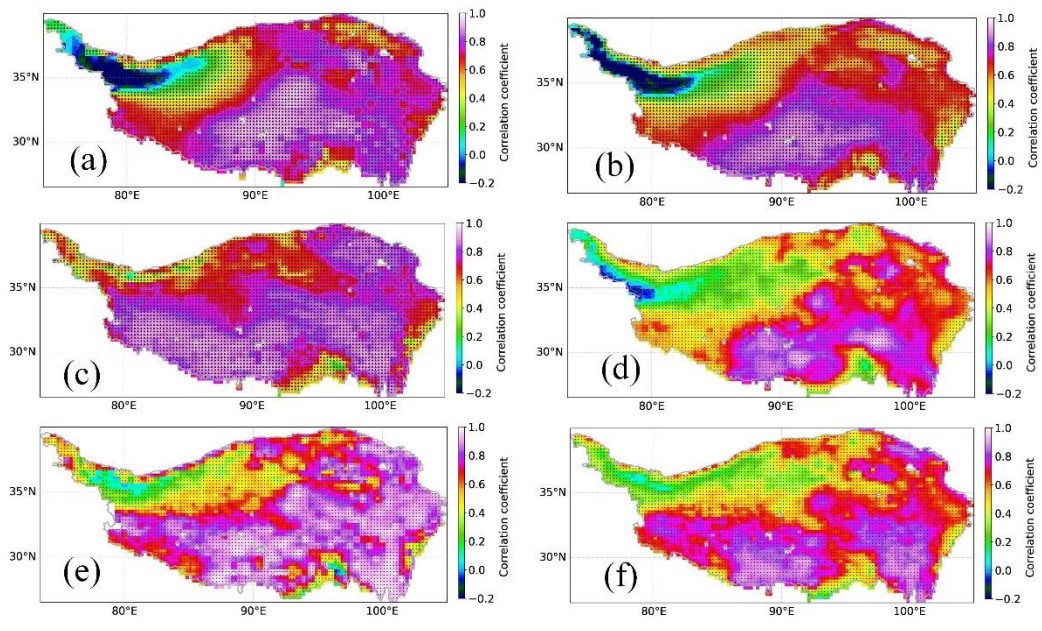

**Figure 4.** Pearson correlation coefficient between the precipitation of (a) CMA and GLDAS, (b) GLDAS and ERA5, (c) CMA and ERA5, (d) GLDAS and CMFD, (e) CMA and CMFD, (f) ERA5 and CMFD over summer periods of May-September. The black dots denote the significant ($p < 0.05$) correlations.

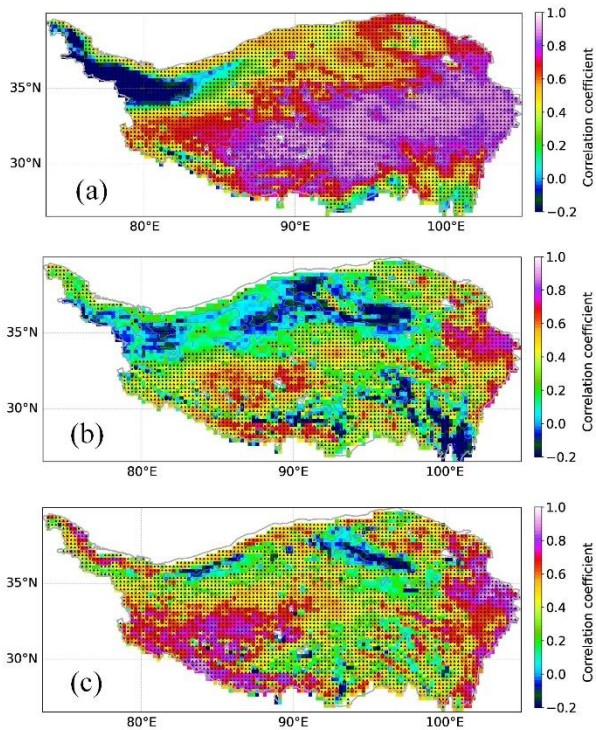

**Figure 5.** Pearson correlation coefficient between the actual evapotranspiration of (a) GLDAS and ERA5, (b) GLDAS and
220 Han ET, and (c) ERA5 and Han ET over summer periods of May-September.

## 4.2 Spatiotemporal patterns of soil moisture droughts

Based on the space-time joint identification approach, 197/193 drought events (consisting of 350/339 drought clusters) were identified on the TP based on the GLDAS/ERA5 SM, which occurred in summer (May–September) from 1961 to 2014. For the identified drought events, drought duration varies from 1 to 5 months, drought area from ~0.4 to ~19 ($\times 10^5$) km$^2$, and
225 drought severity from ~54 to ~6276. Drought area and severity increased synergistically with the increase of drought duration (Fig. S3). Spatially, the accumulated drought duration, severity, and frequency (represented by the number of drought events) presented similar spatial patterns both for the GLDAS and ERA5 SM (Fig. 6). Extensive regions, particularly the central and south TP, endured long lasting (>30 months), comparatively severe (>30), and frequent (>25 events) drought situations. The discrepancy between the GLDAS and ERA5 SM drought is mainly in west TP (west of 85°E). The less severe drought situation
of west TP characterized by ERA5 seems more reliable considering the higher temporal correlation between ERA5 and in situ

SM over the arid NAGARI and the higher consistency between ERA5 and the inter-compared datasets (CMA, CMFD and Han) in Prep and ET than GLDAS over the southwest TP (Fig.3b, Fig. 4, Fig. 5).

To obtain specific knowledge of each identified drought, a schematic map the drought clusters characterizing their location, area, and intensity is given in Fig. 7. Most TP experienced SM droughts with varying intensity. Large scale drought clusters are mainly concentrated in semi-arid and sub-humid regions, i.e. the wet-dry transition zone (Fig. S4). The main difference for drought cluster distribution between GLDAS and ERA5 is also in areas west of 85°E, which is consistent with that of the drought duration, severity, and frequency (Fig.6, Fig. 7). Although the space–time joint identification could reshape the development process of drought, all the identified drought events with their spatiotemporal dynamic patterns can hardly be explicitly presented one by one. In this case, the most severe drought event based on GLDAS is displayed as an example (Fig. 8). Moreover, the standard anomaly of two predominant SM controlling factors in hydrological cycle, that is Prep and PET, are also present to illustrate their driving forces on SM drought. The Prep and PET monthly anomaly maps were observed 1 month before the drought occurrence, considering the time delaying impacts of Prep and PET on SM drought (Van Loon and Van Lanen, 2012). This drought started in May 1994 in the northwest of the TP, with an area of 16% of the TP. After 1 month growing in place, the drought cluster expanded to the southeast in July and the drought area increased to 52% of the TP. Then, the large drought cluster split into two clusters (small in the north and large in the south) in August, which lasted until September, with 35% of the TP area still under drought before the freezing season. Over the 5-month duration, it swept over 73% of the TP and became the most serious drought over 1961–2014. The initial SM drought was co–driven by the persistent negative Prep anomaly and strong PET in April. Under an arid climate in the northwestern TP, the ET was limited by SM. The abnormal drying SM will induce an abnormally low ET and a high surface temperature increase by enhancing the sensible heat flux outcoming. This feedback process aggravates the drought situation. Thus, the drought cluster grew in place until June. The southeastward expansion of drought was synergistically promoted by Prep and PET as both with significant anomalies occurring in June and July. Likewise, the drought persistence in August and September was contributed by the negative Prep anomaly combined with the strong PET capacity, particularly for the large part in the south TP.

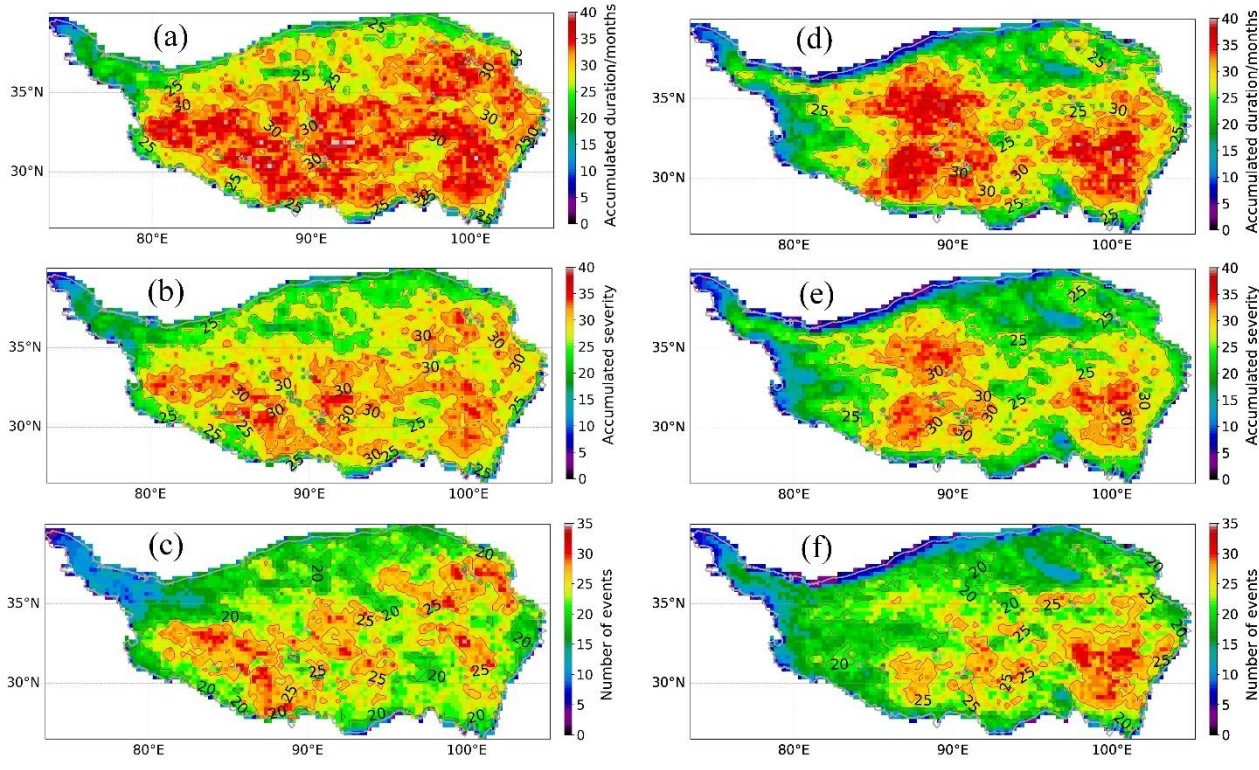

**Figure 6.** Spatial distribution of the accumulated drought duration (a)/(d), severity (b)/(e), and number of events (c)/(f) for the identified soil moisture drought events in summer (May–September) over 1961–2014 in a joint space–time perspective based on GLDAS/ERA5 soil moisture.

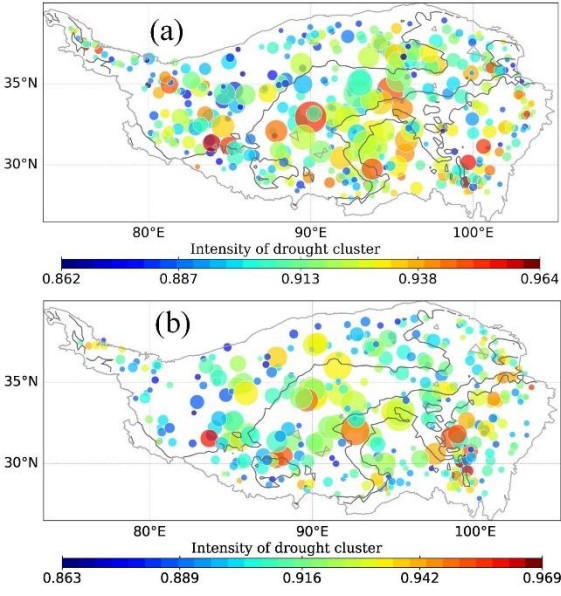

 **Figure 7.** Map of the soil moisture drought clusters for (a) GLDAS and (b) ERA5. The circle centre and size represent the centroid and area of the drought cluster, respectively. The dimgray lines are the dividing lines between different climate zones (the same to Figure S4).

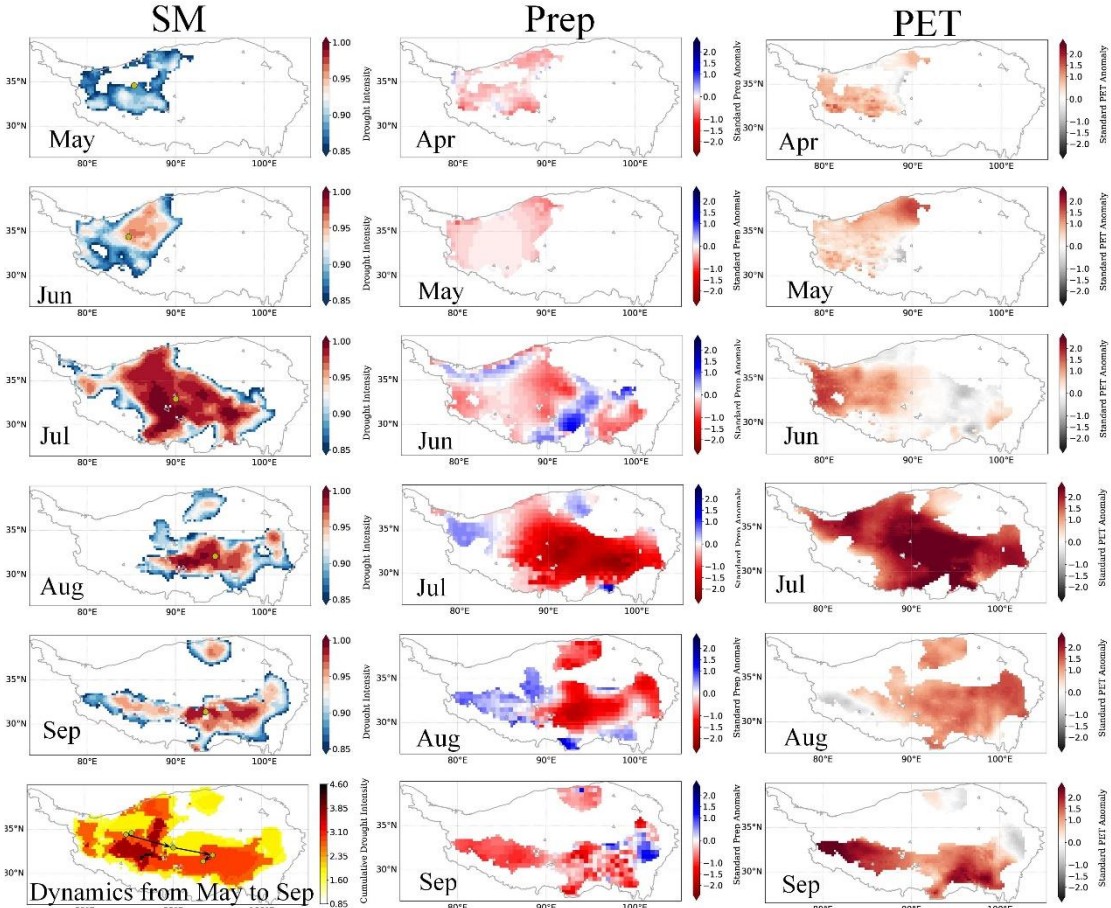

**Figure 8.** Spatiotemporal dynamic patterns of the most severe soil moisture drought event based on GLDAS over 1961–2014 (the left column), the corresponding monthly standard anomaly maps of Prep (the middle column), and PET (the right column) before (1 month) of the drought duration over the drought cluster affected region. For the subgraph in the left bottom marked by "Dynamics from May to Sep", the yellow dots are the centroids of the drought clusters. The black arrow lines show the migration or evolution direction and trajectory of the drought clusters.

### 4.3 Long–term variations in soil moisture droughts

For the entire TP, SM droughts occurred almost every year over 1961–2014 (Fig. S5). However, annual drought showed a significant shift of alleviation in drought severity based on GLDAS and ERA5 SM (Fig. 9a,b). The abrupt wetting shift not only reflected in the SM drought state, but also in the average SM state, Prep, and ET (Fig.S6). The abrupt shift occurred in

the mid to late 1990s at the significance level of $p < 0.05$ based on the Pettitt diagnosis (Pettitt, 1979) (Fig.S7). Supports of the abrupt wetting of the TP can also be drawn from the researches of Zhou et al. (2022), Sun et al. (2019), and Ma and Zhang, (2022). Figure 9a,b shows that the drought of the TP did not present significant ($p<0.05$) tendency before and after the shift. For further insight into the SM drought variations, two predominant controlling factors of SM dynamics, that is Prep and PET, were incorporated. Prep showed an abrupt, substantial increase after the mid to late 1990s, but the change in PET was nonsignificant (Fig. 9c,d,e). It indicates that Prep likely dominated the wetting shift of SM. This is also supported by the significant and relatively higher temporal consistency between the annual SM drought severity and Prep (with the partial correlation coefficient $pr = -0.6$ and $-0.49$) than PET ($pr = 0.48$ and $0.42$) for GLDAS and ERA5. Nevertheless, the slightly enhancing SM drought after the mid to late 1990s should be mainly contributed by the increasing PET as the rising Prep would adversely alleviate the drying tendency. The partial correlation statistics also illustrate that the SM drought trend after the mid to late 1990s is mainly impacted by PET, because the $pr$ between SM drought and PET is 0.59 and 0.48, and the $pr$ between SM drought and Prep is $-0.37$ and $-0.36$ for GLDAS and ERA5, respectively.

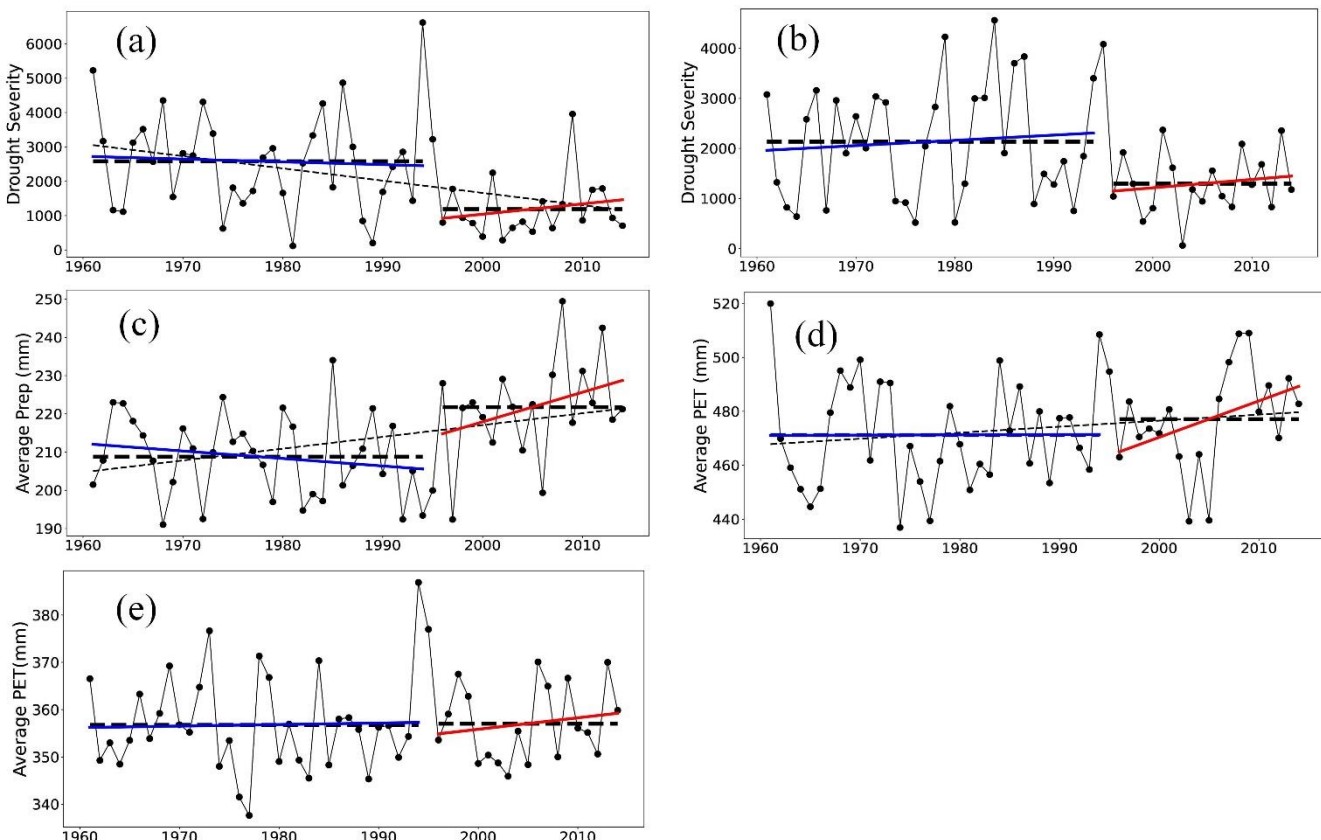

**Figure 9.** Variations of the total annual SM drought severity for (a) GLDAS and (b) ERA5, (c) yearly average precipitation (Prep) of CMA, and potential evapotranspiration (PET) of (d) GLDAS and (e) ERA5 over summer periods (May–September)

from 1961 to 2014. The dotted black lines are the average values over periods before and after the mid to late 1990s. The blue and red lines indicate the tendency for periods before and after the mid to late 1990s, respectively.

To have specific knowledge of the wetting shift of SM drought in space, the difference between the mean SM percentile after and before the mid to late 1990s are considered, together with Prep and PET (Fig. 10). The drought decrease (indicated by the increased SM percentile) occurs in most (~87% for GLDAS and ~83% for ERA5) of the TP and the increase is in small parts of the edge-region (Fig. 10a, b). The major wetting shift toward drought alleviation is mainly distributed in the central TP for GLDAS (east of the black rectangle in Fig.10a), particularly for the semi-arid and sub-humid regions (Fig. S8a).

Nevertheless, the major wetting shift is largely in the interior TP for ERA5 (west of the black rectangle in Fig. 10b), particularly for the arid and semi-arid regions (Fig. S8b). The robust enhancement of SM drought is prevalent in a small part of southeast TP for GLDAS (south of red rectangle in Fig.10a) and northeast TP for ERA5 (north of red rectangle in Fig.10a) with a humid climate (Fig. S8). Moreover, the wetting shift in central TP indicated by GLDAS is partly supported by the greening of this region (Fig. 11). However, the magnitude of wetting need to be further discussed, because the greening of vegetation in the

central TP is not prominent and the greening is jointly impacted by soil moisture and temperature with temperature dominated in the east TP (Fig. 12). The wetting shift in the interior TP indicated by ERA5 is supported by the rapidly expanded lakes in the interior TP after mid to late 1990s  and the greening in the west TP largely controlled by SM (Fig. 11, 12) (Zhou et al., 2021; Yang et al., 2014). Additionally, the drought enhancement of SM for the southeast TP in GLDAS is supported by the browning of vegetation under warming climate (Fig. 11). The abrupt drying of SM for the northeast TP in ERA5 need further

research considering the wetting and the increasing Temp as well as the leading role of Temp in vegetation variation  over this region (Fig. 12).

    The wetting of SM for major TP should be dominated by Prep, in particular for the interior and central-west TP (Fig.10c). The different magnitudes of wetting shift in space, particular in the interior and central-east TP indicated by GLDAS and ERA5 are still impacted by PET (Fig, 10 d,e). Specifically, the decreased PET in central-east promoted the significant wetting

shift in GLDAS, while the decreased PET in west TP contributed the significant wetting shift of the interior TP in ERA5. However, the substantial difference of GLDAS and ERA5 in the spatial pattern of PET discrepancy indicates large uncertainty in PET estimation. For the central and west TP with an arid, semi-arid, and sub-humid climate, SM strongly constrains evapotranspiration variability. Large scale significant wetting shift of SM tends to accelerate the wetting and slow down the warming up of the plateau by increasing the ET and decreasing the surface air temperature. The wetting shift may also produce

impacts on the atmospheric circulation over the TP via land-atmosphere coupling. In addition, the robust enhancement of SM drought in the southeast TP is likely promoted in collaboration by the significant decline of Prep and the substantial increase of PET (red rectangle in Fig. 10c, d). The synergistically abrupt change for Prep and PET toward dryness may pose risks to the transition of the long-term climate state of this region.

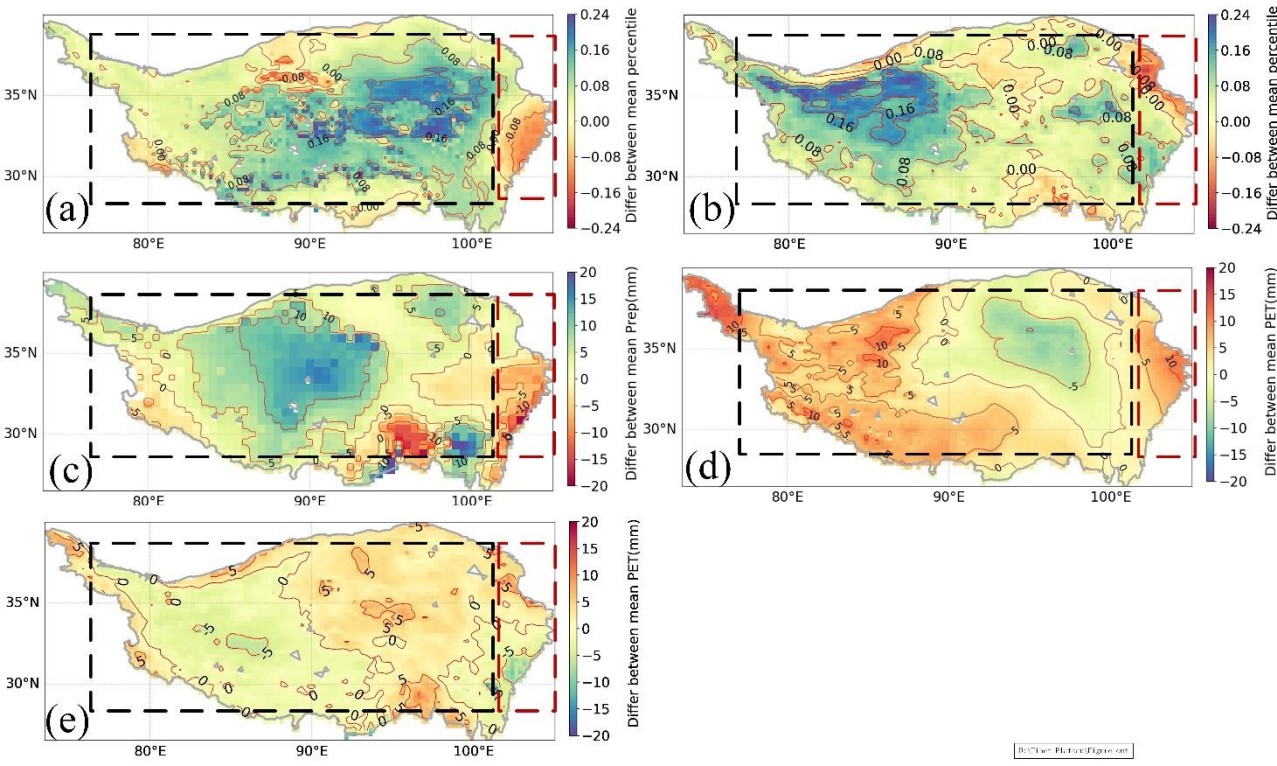

**Figure 10.** Difference (Differ) of the average state of the SM drought for (a) GLDAS and (b) ERA5, mean Prep (c), and mean PET for (d) GLDAS and (e) ERA5 between the periods after and before the mid to late 1990s.

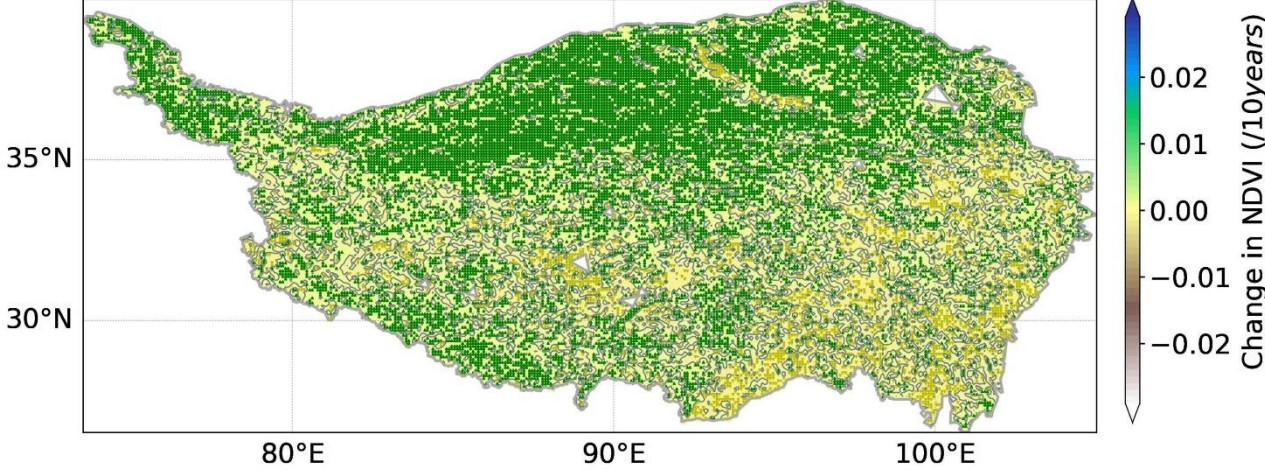

**Figure 11.** Variations of NDVI over summer periods of May–September from 1982 to 2015 for the Tibetan Plateau. The greed/brown △/▽ denotes the significant ($p < 0.05$) increasing/decreasing trends.

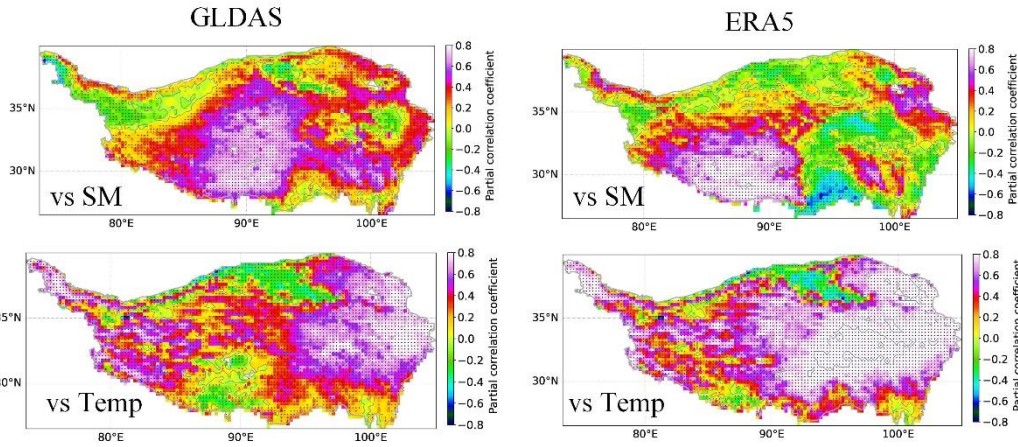


**Figure 12.** Spatial distribution of the partial correlations between the NDVI and soil moisture (SM)/surface air temperature (Temp) over summer periods of May–September from 1982 to 2015 for GLDAS and ERA5, respectively. The black dots denote the significant correlations.

After the abrupt wetting shift in the mid to late 1990s, the correlation between SM drought and Prep anomaly decreased,
while that with PET increased for most TP, in particular for the west plateau (Fig.S9). This suggests that after the shift in SM, the Prep (i.e., water) controlling capacity for variations in SM drought was weakened, while the PET (i.e., energy) control was intensified. This phenomenon may indicate a climate regime changing tendency of wetting over the TP. Due to the wetting shift and the associated water and energy control adjustment, the trend characteristic of SM drought was analyzed separately before and after the shift. Generally, SM drought is increasing (with decreasing SM percentile) in the southwest half of the
plateau and decreasing in the northeast half in both stages based on GLDAS and ERA5 (Fig.13a1-b2). This feature is supported by the trend pattern of both Prep and PET, in particular for PET over later stage (Fig.13c1-e2). Moreover, the later stage presents more evident wetting trends with a larger magnitude of SM variations and an expanded area with significant changes than the prior stage (Fig.13a1-b2). The wetting trends in the northeast half are likely contributed by the combing of Prep and PET. The primary difference between GLDAS and ERA5 is in the west TP (the region with red rectangle). The significant
drying was detected by GLDAS in the later stage, while observed by ERA5 in the prior stage. The SM drying in the west TP seems dominated by PET, with the dominance strengthened in the later stage (Fig. 13 b2,d2,e2). This is also supported by the significant ($p < 0.05$) and high correlations of SM with PET, but the un-significant ($p > 0.05$) and low correlations with Prep in west TP (Fig.S9). However, the PET estimation in the west TP tend to bear large uncertainty due to the scarce in situ measurements there. The climate drying or wetting of the west TP needs to be further discussed. In addition, the significant
drying trends in the southeast TP (emerald rectangle) are pronounced, which is likely controlled by Prep and PET in collaboration.

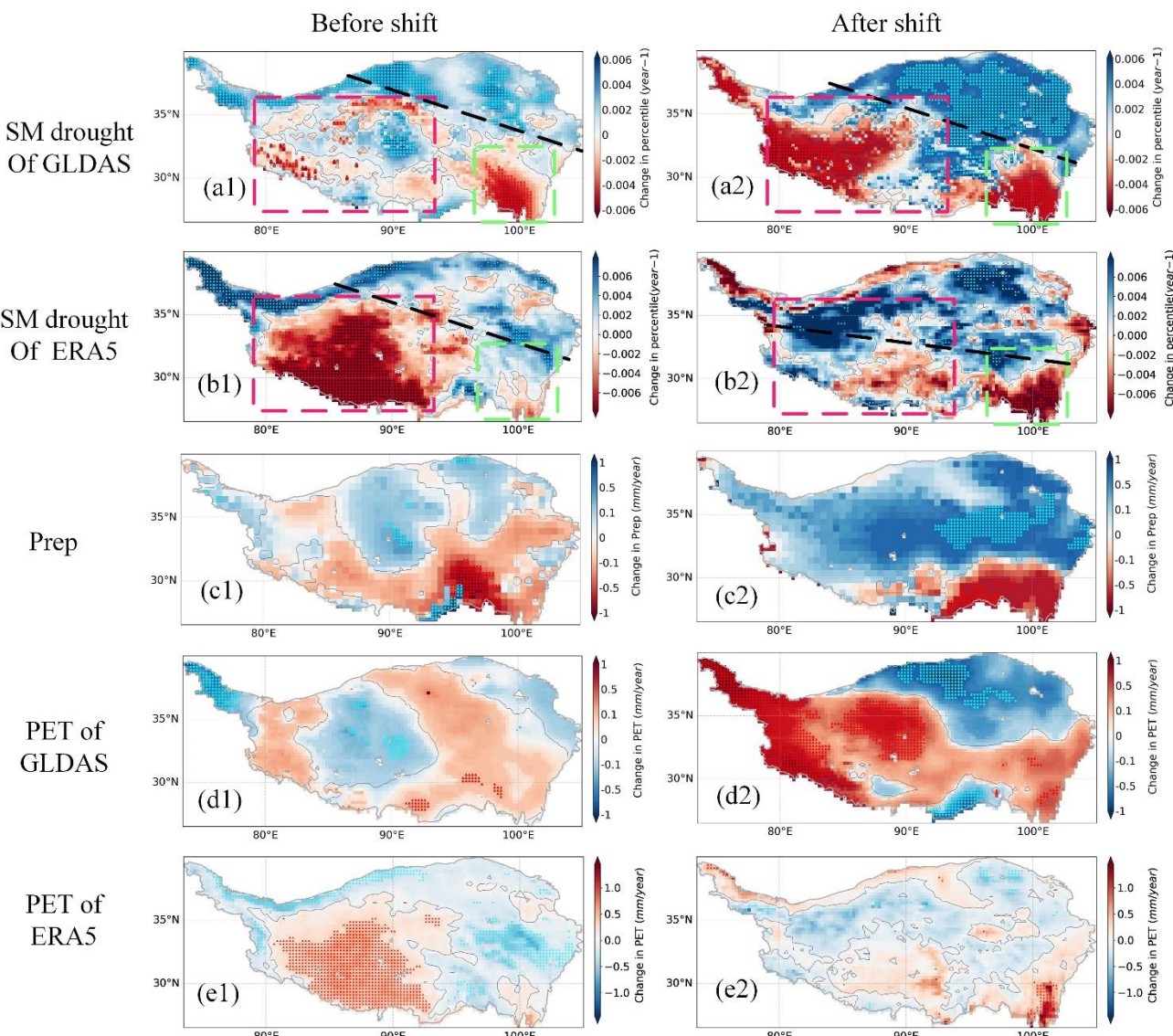

**Figure 13.** Spatial distribution of the trends in soil moisture (SM) percentile, precipitation (Prep), and potential evapotranspiration (PET) before and after the wetting shift. Note that the $\triangle/\triangledown$ with cyan/red denotes the significant ($p < 0.05$) increasing/decreasing trends.

## 5 Discussion

### 5.1 Possible linkages to the soil moisture drought shift

The SM droughts of the TP in summer (May–September) had a significant shift of alleviation in the mid to late 1990s. Our results demonstrated that Prep was the primary driver of the shift and PET the secondary. Prep variability on an inter-annual scale for summer TP is controlled by remote moisture transport driven by the general circulation of the atmosphere (Wang et al., 2017; Li Ying et al., 2019). In summer, the TP is under the control of the ISM and westerlies, with limited influence from the EASM (Yao et al., 2012). The westerly jet stream with its position and strength can be reflected by the AMO because of their high correlations on the inter-decadal scale (Sun et al., 2020). The ISM and EASM with their strength can be largely manifested in the PDO, AMO, and ENSO due to their close relationships on the chronological scale. Considering the fact that the global sea-surface temperature can be partly impacted by solar cycles, solar cycles were also incorporated in the analysis of its influences on the circulations and Prep of the TP.

The inter-decadal correlation analysis of the aforementioned circulation and monsoon indexes suggests that the EASM is significantly ($p<0.05$) related to the AMO, PDO, and ENSO. The ISM is significantly correlated with the AMO and PDO. The correlation analysis based on 11-year running average statistics shows that solar circles have significant impacts on the ENSO and ISM. Due to the complex cross-impacts among the oscillation and circulation indexes and the subsequent multiple impacts on Prep, the partial correlation between these indexes and Prep were analysed in space. The results show that the inter-decadal variations of summer Prep for the plateau are largely impacted by the AMO and PDO, while locally impacted by the ENSO (Fig. 14 a,b,c). The ISM has significant positive impacts on the Prep of south TP, while negative impacts on the Prep of north TP. The EASM has positive impacts on the Prep of southeast TP, while negative influences on the Prep of central north TP (Fig. 14 e,f). Additionally, the solar cycles highly impacted Prep located in parts of the west and southwest plateau (Fig. 14 d).

The AMO has experienced a phase transition from cold (negative) to warm (positive) since the mid-1990s (Fig. 15a). In the warm phase, the AMO would induce a wave train of cyclonic and anticyclonic anomalies over Eurasia in summer (Fig.16). These anticyclonic anomalies to the east of the TP tend to cause east wind anomalies, that is, the weakening of the westerly winds at 200hPa near the TP (Fig.16, gray rectangle). The weakened westerlies would reduce the water vapor transport beyond the eastern TP boundary and facilitate the convergence of water vapor (i.e., Prep formation) over the interior and central TP (Wang et al., 2017; Zhou et al., 2019). The more the moisture gathered, the more the Prep falls in this region. At the same time, the abnormal cyclone to the west of the TP promoted the northward and eastward water vapor transport imported from the southwest boundary of the TP (Fig. 16, the green rectangle), which also increased the wetting shift of the interior TP (Zhang et al., 2019b; Sun et al., 2020). On the other hand, the phase transition of AMO from cold to warm is inclined to cause the weakening of the ISM and EASM due to the significant negative correlations between the AMO and the ISM and EASM. This impact seems more pronounced on ISM with significant decreasing tendency after mid-2000s (Figure 15e). The weakened ISM would contribute to the wetting of the north TP, while slow down the wetting of central south TP through its impacts on

the water vapor transport from south boundary (Figure 14e). The PDO seems mainly in its cold (negative) phase after the mid

to late 1990s on inter-decadal scale (Fig. 15b). The cold phase of PDO is characterized by a cool wedge of lower than normal

sea-surface ocean temperature in the eastern equatorial Pacific and a warm horseshoe pattern of higher than normal sea-surface

temperature connecting the north, west and southern Pacific. The relatively higher sea-surface temperature is inclined to

weaken the land-sea thermal contrast in summer, thus probably contribute to the weakening of the ISM and EASM (Li et al.,

2010; Zhang and Zhou, 2015). The weakening EASM tends to contribute to the wetting of the interior TP and the drying of

southeast TP (Fig. 14 f). Besides, the ENSO and solar cycles are probably not the main factors driving the wetting shift of the

TP as no evident phase difference between the periods before and after mid to late 1990s present in the nino3.4 and SRF

indices (Fig. 15 c,d). In addition, the supportive impacts of PET on the wetting shift of the central-east in GLDAS and the west

in ERA5 largely benefited from the decreased downward solar radiation (solar dimming) and wind speed (wind stilling) (Fig.

10b,c,e,f).

Overall, the shift variation in SM drought is likely predominantly driven by the phase transition of the AMO from cold

to warm since the mid-1990s. It should also be related to the phase transition of the PDO and the weakening ISM and EASM

dominated by the ocean oscillation. Additionally, the solar dimming and wind stilling are also contributors of the wetting shift.

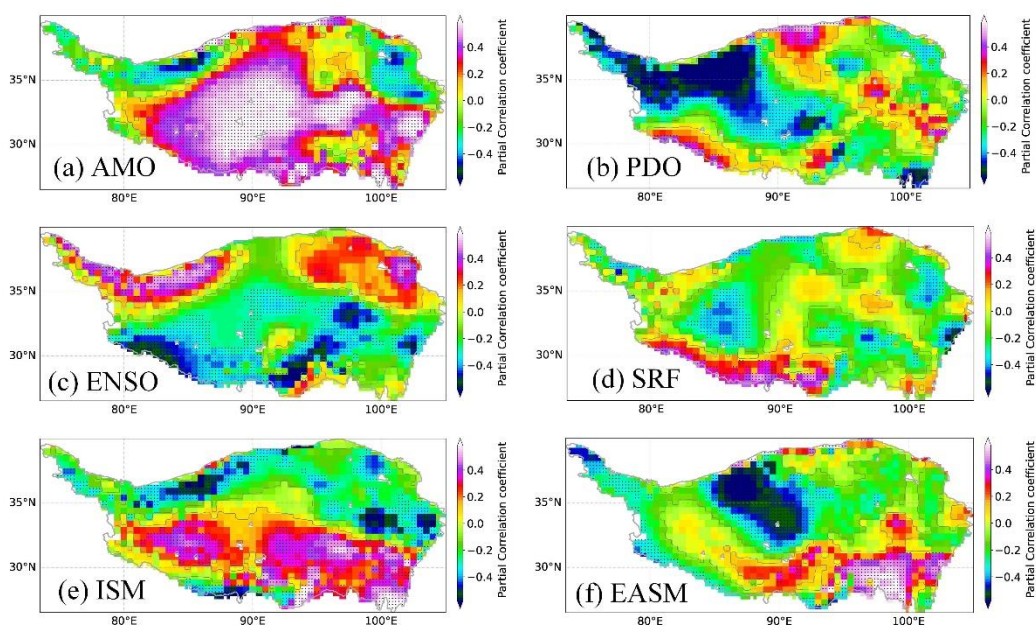

**Figure 14.** Spatial distribution of the partial correlation coefficients between the precipitation and the (a) AMO, (b) PDO, (c)
ENSO (nino 3.4), (d) SRF, (e) EASM and (f) ISM index on inter-decadal scale.

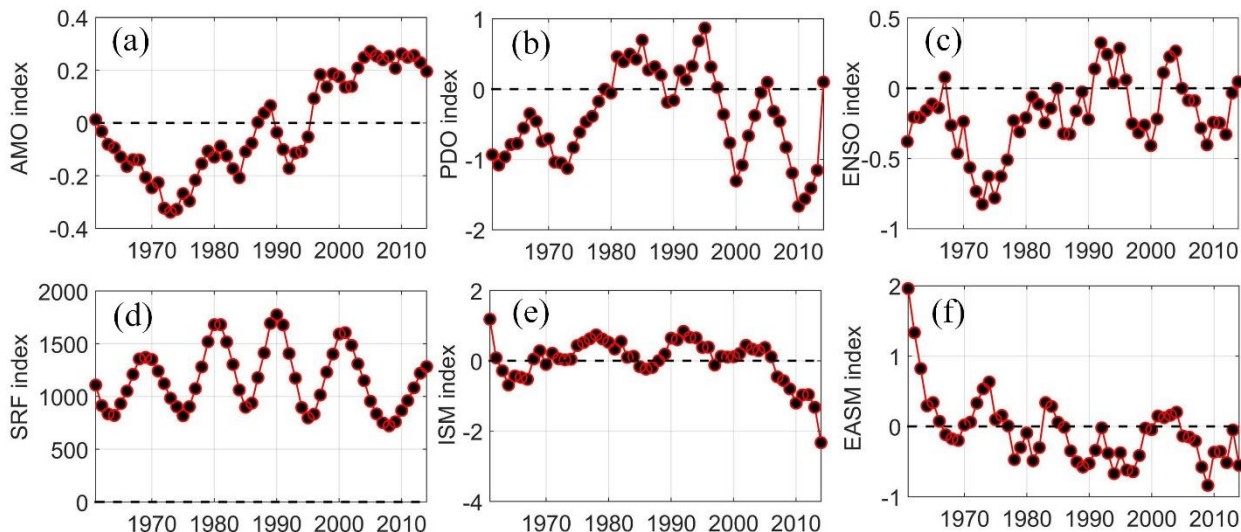

**Figure 15.** Inter decadal variations of the (a) AMO, (b) PDO, (c) ENSO (nino 3.4), (d) SRF, (e) ISM, and (f) EASM index for June–July–August (JJA) over 1961–2014.

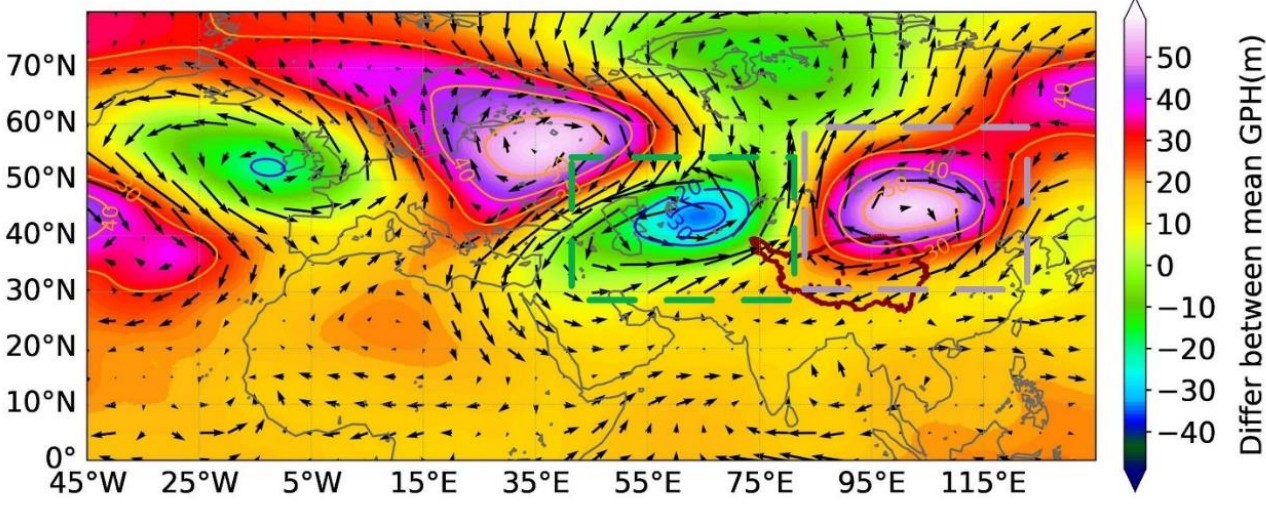


**Figure 16.** Differences (Differ) in summer (July) geopotential height (GPH) at 200 hPa between the warm (1995–2014) and cold (1961–1994) phases of the AMO (the later stage minus the former stage). The bold red line represents the border of the TP.

## 5.2 Potential drivers for the spatial pattern of drought trends

Due to the abrupt wetting shift of SM in the mid to late 1990s, its water and energy control capability had substantial change in space over the plateau. The SM drought trends presented differences before and after the shift, with more significant

wetting in the later stage. The significant wetting of SM in the northeast TP was synergistically controlled by the increasing Prep and decreasing PET (Section 4.3). The Prep increase is related to the positive westerly wind anomalies over the entrance of the East Asia westerly jet in the upper troposphere and the strengthened northward water vapor transport from the south and east (Wang et al., 2018; Zhang et al., 2019b). The PET decrease is probably resulted from the decline in Radi, wind speed, and VPD (Fig. S11). Wind stilling is considered to be the factor with more importance by Zhang et al. (2009). The significant drying SM after the mid to late1990s in the western TP dominated by the increasing PET based on GLDAS is likely not a true reflection of SM. Because the PET based on the meteorological stations was decreasing in Yin et al., 2013 and Yang et al., 2014. In addition, the significant drying in the southeast TP was a combined effect of the increasing PET and decreasing Prep (Fig. 13). The increasing PET is likely dominated by the considerable increase in Temp for the southeast TP as the increase in windspeed and Radi are generally not supported (Fig. S11) (Zhang et al., 2009; Yang et al., 2014). The decreasing Prep in the southeast is probably related to the weakening of EASM and ISM, in particular for the significant weakening of the ISM (Fig. 14e,f, 15e,f). The weakening ISM may decrease the water exchange between the Asian monsoon region and the plateau (Yang et al., 2014). Thus, this phenomenon led to less Prep and drying SM in the ISM affected southeast region (Fig. 13).

Overall, the substantial wetting in the northeast TP is probably a combing forces of the decreasing Radi, wind speed, VPD and the increasing rainfall input from the west and south. However, the significant drying in southeast TP is largely dominated by the increasing Temp and the decreasing rainfall input dominated by the weakening monsoon of the ISM under global warming.

## 5.3 Implications of the abrupt wetting shift of SM on TP climate

The interior and central TP experienced an abrupt wetting shift in SM over summer (May–September) in the mid to late 1990s, particularly for the arid and semi-arid zones. The SM is generally dominated by water and energy from the atmosphere. At the same time, as the largest storage component of Prep and radiation anomalies on land, SM also controls the water and energy feedback from the land surface to the atmosphere by coupling with ET. The shift variations in SM tend to produce changes to the local or regional underlying incoming water and energy redistribution, that is, changes in the SM regime (Koster et al., 2016).

The SM regime is often distinguished based on the sensitivity of evaporation to SM change, which can be reflected in the relationship between SM and evaporative ratio (the fraction of latent heat in net radiation) (Koster et al., 2011). Figure 17 compares this relationship before and after the wetting shift for the TP. The sensitivity of evaporation to SM change significantly increased after the wetting shift as the slope of the probability distribution increased substantially for GLDAS and the centre of most probability has entered the wet end for ERA5. In the arid and semi-arid climate dominated TP, the SM is relatively dry on average and the evaporation is low. The limited SM cannot become too much lower during dry years. By contrast, it is more likely to produce impacts on evaporation and the overlying atmosphere through the surface energy budget during wet years. With the wetting shift in the mid to late 1990s, the SM seems to have entered a regime where it can have significant impacts on evaporation and the overlying atmosphere.

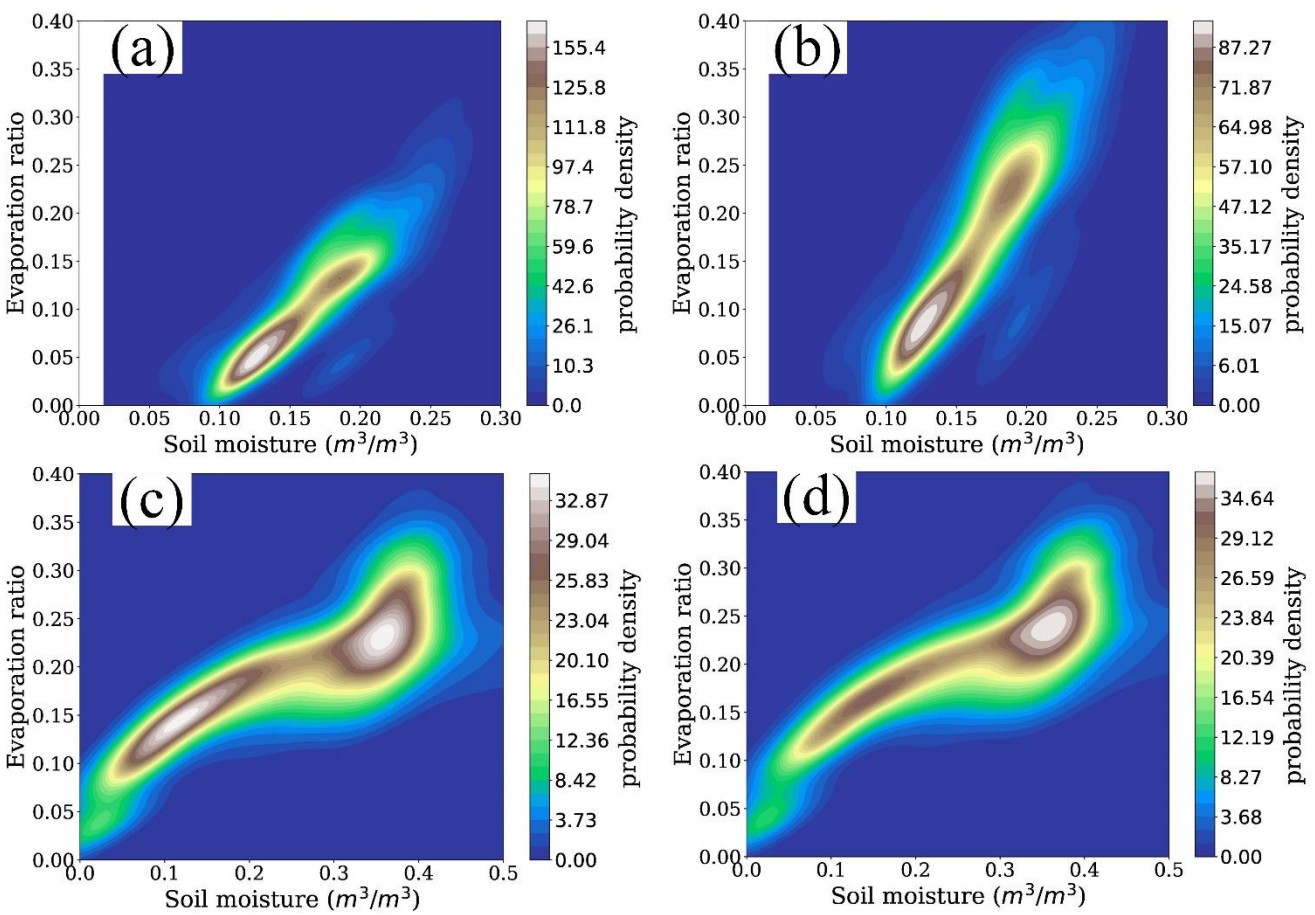

**Figure 17.** Probability distribution of the relationship between the evaporative ratio and the soil moisture in the stages before (a)/(c) and after (b)/(d) the mid to late 1990 for the TP based on GLDAS/ERA5.

SM regime changes may indicate a changing tendency of the climate system. The ratio of long-term average Prep to PET (Prep/PET, i.e., the Aridity index [AI]) provides a notable method of determining the climate regime of a region (Fu and Feng, 2014). The comparison of summer Prep/PET before and after the mid to late 1990s indicates an inclination of phase transition from an arid to semi-arid, semi-arid to sub-humid and sub-humid to humid climate in summer over the wet-dry transitional zone of the TP based on GLDAS and ERA5 (Fig. S4), because the dividing lines for both expand to the northwest in the later stage (Fig. 18, 0.5 and 0.65 lines in the orange rectangle). The significance of this change can be reflected in the comparison of the statistics (mean, median, p25 and p75) from the grid-based summer Prep/PET with a varying Aridity index (Fig. S4) between the prior and later stages (Fig. 19). The overwhelmingly higher Prep/PET for the later (pink shaded area) than prior stag over the arid, semi-arid, and sub-humid zones (the region with AI in 0.1-0.65) illustrates the significant wetting of the climate, particular for the arid and semi-arid regions. The plateau enhances the Asian summer monsoon and modulates its

variability by exerting strong thermal forcing on the mid-troposphere over the mid-latitude of the Northern Hemisphere during summer and spring (Yanai et al., 1992). The possible impacts of the abrupt wetting on Asia and even global climate via the strong land-atmosphere coupling over the transitional zone should be attentioned. In addition, climate controls the structure and function of the ecosystem (Chen et al., 2013a). The large impact of the wetting climate transition on the original ecosystem must be considered, especially for parts of the TP with the most sensitive and fragile ecosystem.

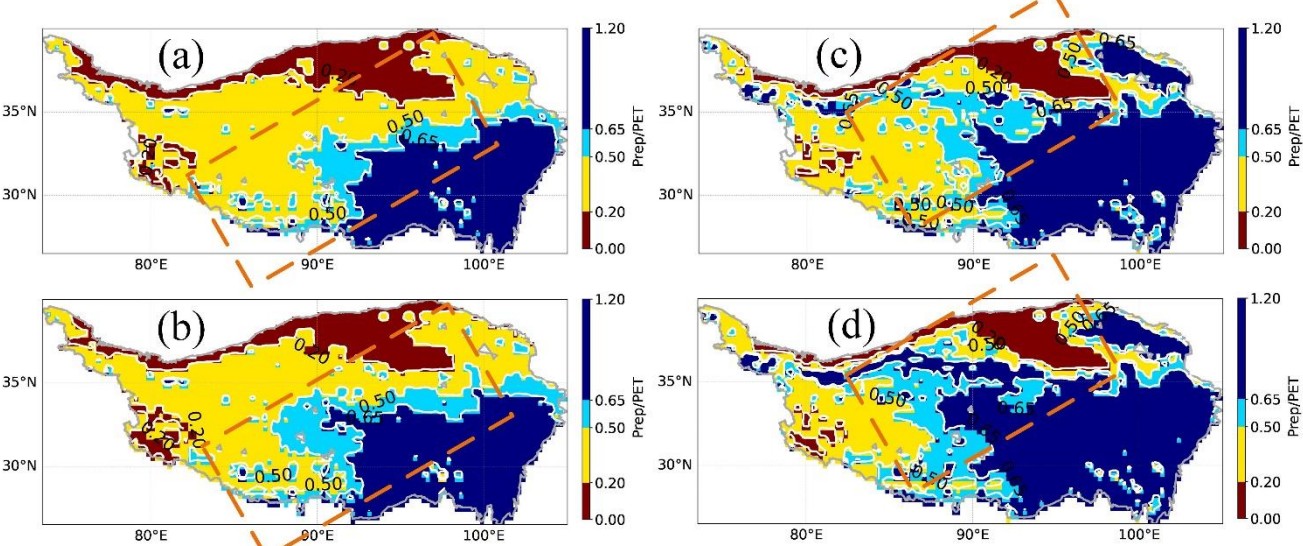

**Figure 18.** Multi-year average Prep/PET for summer periods of May–September over the stages of before (a)/(c) and after (b)/(d) the mid to late 1990s based on GLDAS/ERA5.

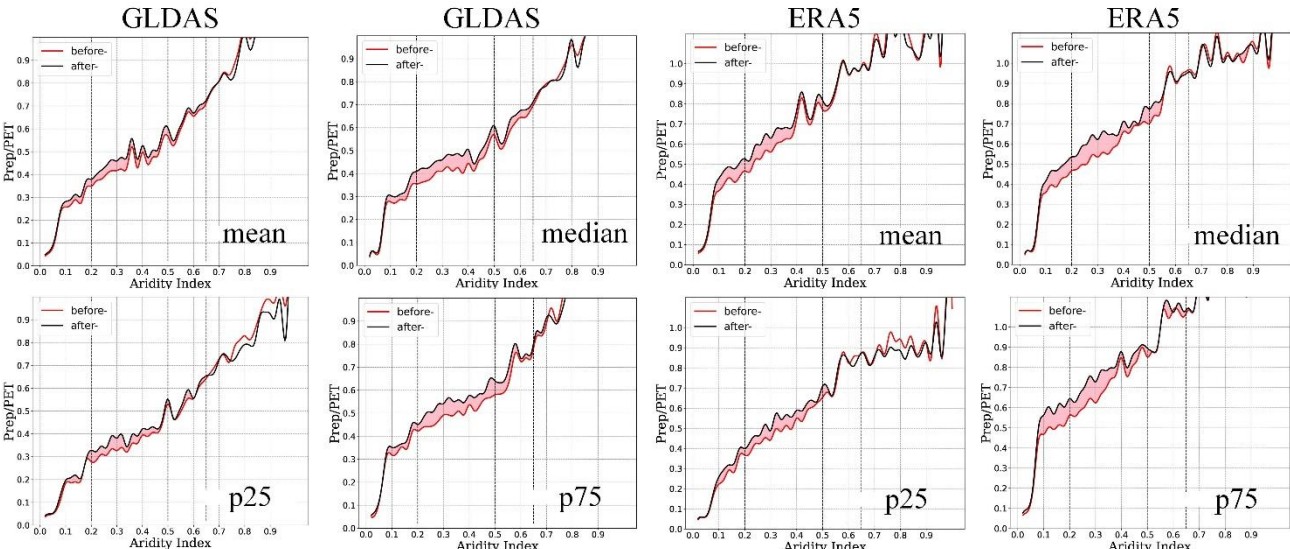

**Figure 19.** Variations of the Prep/PET statistics (mean, median, p25 and p75) over summer periods (May–September) with the Aridity index (AI, Figure S3) for both cases before (before-) and after (after-) the wetting shift based on GLDAS and ERA5. p25 and p75 are 25th and 75th quantile of the cumulative distribution of Prep/PET. The pink shaded areas represent the degree
of wetting shift.

## 6 Conclusion

This study explored the climate wetting/drying of the TP over the last half century from variations of historical SM droughts. The spatiotemporal patterns and long-term variations of summer (May–September) SM droughts and the climate causes were comprehensively investigated based on multi-source observation and reanalysis data. To improve the reliability
of the results, both the GLDAS and ERA5 SM were used based on a comprehensive data validation.

As a result, 197/193 drought events (consisting of 350/339 drought clusters) were identified based on GLDAS/ERA5 SM. Spatially, extensive regions, particularly for the central and south TP endured long-lasting, comparatively severe, and frequent droughts. Large-scale drought clusters are mainly concentrated in the semi-arid and sub-humid regions. The most severe drought event occurred in May 1994 and affected over 73% of the TP over 5 months' duration, driven by persistent, abnormally
low Prep along with strong PET.

The TP experienced a significant phase shift of SM drought alleviation in the mid to late 1990s, indicating an abrupt hydro-climate shift to a wetter plateau. Drought alleviation occurred in ~87% and ~83% of the plateau based on GLDAS and ERA5, respectively. The major wetting shift for ERA5 is in the interior TP, located in the arid and semi-arid regions. Nevertheless, the significant wetting shift for GLDAS is in the central TP, distributed in the semi-arid and sub-humid regions.
The substantial wetting of the interior plateau is likely more credible considering the higher temporal consistency of the ERA5 and in situ SM over the arid and semi-arid regions and the higher correlation between the ERA5 and the inter-compared datasets (CMA, CMFD and Han) in Prep and ET than GLDAS over the southwest plateau. Although the wetting shift of central TP in GLDAS is partly supported by the greening of vegetation, the magnitude of wetting needs further investigation. Before and after the phase shift, the SM drought did not present significant trends for the entire TP. Spatially, the SM wetting in the
northeast half of the plateau became more prominent after the shift as a larger magnitude of SM wetting and an expanded area with significant changes are observed in the later stage.

We demonstrated that the shift alleviation of SM drought was predominately controlled by its dominant input (i.e., Prep), while its dominant output driver (i.e., PET) was a secondary contributor. By contrast, the drought trends of SM were combing forces of Prep and PET, the driving forces of PET increased after the wetting shift in the mid to late 1990s. Furthermore, the
Prep driving on the shift of SM drought can be largely attributed to the phase transition of the AMO from cold to warm accompanied by the weakening westerly since the mid-1990s, and partly to the phase transition of the PDO and the weakening ISM and EASM dominated by the ocean oscillation. The PET effects on wetting shift should be mainly benefitted from the solar dimming and wind stilling, which offsets the considerable increase in Temp. By contrast, the PET impacts on the wetting

trends are likely combing forces of solar radiation, wind speed, and vapour pressure deficit. The Prep impacts on trending tend
to be related to the weakening ISM under global warming.

      Overall, with the objective of exploring the climate wetting/drying of the TP under the background of profound global climate change over the last half century, this study provides a comprehensive investigation on SM droughts of the TP. According to our review of the literature, the core variable of land-atmosphere water and energy cycle, that is SM, was first adopted as an indicator to characterize the drought and indicate the climate wetting/drying of the TP. A significant wetting
shift in the plateau was observed. Furthermore, the wetting shift was interpreted synthetically from not only the dominant climate-controlling factors of SM but also the general circulation of the atmosphere. We demonstrated that the wetting shift indicated a SM regime change and an inclination of phase transition from arid to semi-arid, semi-arid to sub-humid, and sub-humid to humid climate. The large impact of the climate transition on the original sensitive and fragile ecosystem of the TP should be considered. The possible impacts of the overall wetting shift on global climate through the strong land-atmosphere
coupling should also be attentioned. The aforementioned innovative work provides evidence for an abrupt wetting shift of the plateau over summer periods and is significant for further understanding climate change of the TP.

**Code and Data availability**

      GLDASv2.0/Noah dataset is available from https://disc.gsfc.nasa.gov/datasets/GLDAS_NOAH025_M_2.0/summary?keywords=GLDAS. Precipitation and surface
temperature data provided by the CMA are available at http://www.nmic.cn/data/cdcindex/cid/6d1b5efbdcbf9a58.html. ISMN data is available from https://ismn.geo.tuwien.ac.at/en/. The AMO indice is available at http://www.psl.noaa.gov/data/timeseries/AMO/, and the PDO indice is at https://www.ncdc.noaa.gov/teleconnections/pdo/. The ISM index is available from http://apdrc.soest.hawaii.edu/projects/monsoon/seasonal-monidx.html, and EASM index is from http://ljp.gcess.cn/dct/page/65540. Niño 3.4 index is available from
https://psl.noaa.gov/data/timeseries/monthly/NINO34/. SRF index is available at http://www.esrl.noaa.gov/psd/data/correlation/solar.data. The ERA5 dataset is available at https://cds.climate.copernicus.eu/cdsapp#!/search?type=dataset. Further datasets and code can be accessed upon request from the corresponding author.

**Author contributions**

Yongwei Liu and Yuanbo Liu conceived and designed the research. Yongwei Liu performed the experiments, wrote, and edited the paper. Wen Wang helped revised the paper. Han Zhou and Lide Tian gave constructive suggestions to improve the paper.

**Conflict of Interest statement**

The authors declare that they have no known competing financial interests or personal relationships that could have
appeared to influence the work reported in this paper.

**Acknowledgements**

This work was supported by the Second Tibetan Plateau Scientific Expedition and Research Program (2019QZKK0202), the National Natural Science Foundation of China (41901049, 41430855, 41961134003), Jiangsu Science and Technology Planning Youth Project (BK20191097) and National Key R&D Program of China (2018YFE0105900).

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
