# Peer review of "Historical droughts manifest an abrupt shift to a wetter Tibetan Plateau"

_Hydrology and Earth System Sciences, 2022_

## Referee Comment (RC1)

**Comments on "Historical droughts manifest an abrupt shift to a wetter Tibetan Plateau" by Liu et al.**

This study analyzed the climate wetting/drying of the Tibetan Plateau from variations of historical soil moisture droughts over 1961-2014, focusing on the spatiotemporal patterns, long-term variations of soil moisture, and the related climate causes of summer (May–September). Multiple observation and reanalysis data were used for the analysis. To this reviewer, these analyses are very important to quantify the various aspects of the soil moisture droughts and their causes.

In reading through the manuscript, despite that fact that the authors have done a lots analyses, there seem some important technical issues that need to be considered carefully to ensure that the results of the study are reliable. Some of these issues are listed as follows:

1. The consistency of the used variables used in describing the water components. The study used different data sources for the analysis, e.g. the soil moisture is extracted from the GLDASv2.0/Noah dataset from the depth of 0-10cm, precipitation and air temperature from interpolation the gauge data provided by the Chinese Meteorological Administration, potential evapotranspiration, wind speed, radiation, vapour pressure deficit, latent heat flux, and net radiation flux again from the output and forcing datasets of GLDASv2.0/Noah. The authors need to consider if the relevant quantities are consistent before carrying out further analysis. Inconsistency in precipitation in the CMA data and that in GLDASv2.0/Noah can cause lots of inconsistencies in the derived soil moisture and its relationships to precipitation. See e.g. https://journals.ametsoc.org/view/journals/bams/99/2/bams-d-16-0074.1.xml for how to verify and ensure the consistency of the climate data records.

2. GLDASv2.0/Noah data is strictly not a climate data record, the authors need to verify the temporal consistency of the used variables to make sure that the trends in the used variables are true reflection of the actual states of Tibetan Plateau and not caused by e.g. the change of the forcing data. A suggestion is compare the relevant variables with the ERA5 data and discuss the uncertainties. Some comparison to in-situ observation should also help to ensure the validity of the conclusions. Examples of such comparisons could be for precipitation (e.g. those from CMA and input to GLDASv2.0/Noah), soil moisture and evaporation from in-situ observation and remote sensing, e.g. https://www.mdpi.com/2072-4292/13/18/3661/htm; https://www.mdpi.com/2072-4292/12/3/509, https://essd.copernicus.org/articles/13/3513/2021, among many others.

3. Changes in vegetation coverage is closely related to the changes in soil moisture and temperature, this seems completely neglected by the authors. See e.g. https://link.springer.com/article/10.1007/s10584-009-9787-8.

4. Figure 4 is rather difficult to comprehend, perhaps a Hovmöller diagram is more effective.

5. Trend lines and designated changes in Figures 5, 9 and 11 appear arbitrary, unless the authors can provide more details in trend detection that identifies 1995 as the year of abrupt change. Such a change in Fig. 5 is also not observed nor explained by the monsoon indices, AMO, PDO and ISM. These relationships should be explored more in detail. Does ENSO have any impact on the precipitation and circulation patterns? How about the solar cycles?

---

## Author Response (AR1)

**Reply to the comments from the reviewers**

We appreciate the constructive criticisms and suggestions. We have addressed the main changes and each of their concerns below.

A summary of the major changes:

(1) The ERA5 dataset was incorporated in the analysis of the climate wetting/drying of the Tibetan Plateau. The performance of ERA5 and GLDAS datasets were inter-compared and the uncertainties were discussed in the updated manuscript.

(2) The in-situ soil moisture observations from the ISMN were used to verify the validity of the GLDAS and ERA5 soil moisture. Additionally, the precipitation from the China Meteorological Forcing Dataset (CMFD) and the actual evapotranspiration developed by Han et al. (2020, 2021) were incorporated in the precipitation and actual evapotranspiration inter-comparisons to improve the reliability of the results.

(3) The wetting shift was investigated more in detail and its possible linkages were further explored in section 4.3 and 5.1 in the updated manuscript.

==============================

**Comments from Reviewers:**

**Reviewer Comments 1:**

**Comments on "Historical droughts manifest an abrupt shift to a wetter Tibetan Plateau" by Liu et al.**

This study analyzed the climate wetting/drying of the Tibetan Plateau from variations of historical soil moisture droughts over 1961-2014, focusing on the spatiotemporal patterns, long-term variations of soil moisture, and the related climate causes of summer (May–September). Multiple observation and reanalysis data were used for the analysis. To this reviewer, these analyses are very important to quantify the various aspects of the soil moisture droughts and their causes. In reading through the manuscript, despite that fact that the authors have done a lots analyses, there seem some important technical issues that need to be considered carefully to ensure that the results of the study are reliable. Some of these issues are listed as follows:

1. The consistency of the used variables used in describing the water components. The study used different data sources for the analysis, e.g. the soil moisture is extracted from the GLDASv2.0/Noah dataset from the depth of 0-10cm, precipitation and air temperature from interpolation the gauge data provided by the Chinese Meteorological Administration, potential evapotranspiration, wind speed, radiation, vapour pressure deficit, latent heat flux, and net radiation flux again from the output and forcing datasets of GLDASv2.0/Noah. The authors need to consider if the relevant quantities are consistent before carrying out further analysis. Inconsistency in precipitation in the CMA data and that in GLDASv2.0/Noah can cause lots of inconsistencies in the derived soil moisture and its relationships to precipitation. See e.g. https://journals.ametsoc.org/view/journals/bams/99/2/bams-d-16-0074.1.xml for how to verify and ensure the consistency of the climate data records.

**Response 1:** Done. The consistency of precipitation and temperature from the gauge interpolated data provided by the Chinese Meteorological Administration (CMA) and the GLDAS forcing input data were evaluated. The precipitation and temperature of ERA5 and China Meteorological Forcing Dataset (CMFD) were incorporated for inter-comparison (He et al., 2020). The temporal correlations between different datasets were analyzed. The results show that the precipitation from both CMA, GLDAS, ERA5 and CMFD present higher temporal consistency in the south and east than the north and west Tibetan Plateau (TP) (Fig. 4). The precipitation of CMA and GLDAS has good consistency in large of east and south TP, but low consistency in the northwest area (Fig.4a). The precipitation of CMA presents better consistency with ERA5 and CMFD than GLDAS (Fig. 4 b,c,d,e). Likewise, the precipitation of ERA5 has better temporal consistency with CMA and CMFD than GLDAS, in particular for the southwest TP (Fig. 4 a,c,d,f). The best consistency of precipitation is between CMA and ERA5 (Fig. 4c). In terms of the surface air temperature, both CMA, GLDAS, ERA5 and CMFD show quite high temporal consistency. The correlation coefficient between any two datasets is over 0.8 for the vast majority of the TP (Figure R1). (lines 196-201 in the updated manuscript)

[Figure]

Figure 4. Pearson correlation coefficient between the precipitation of (a) CMA and GLDAS, (b) GLDAS and ERA5, (c) CMA and ERA5, (d) GLDAS and CMFD, (e) CMA and CMFD, (f) ERA5 and CMFD over summer periods of May-September. The black dots denote the significant ($p < 0.05$) correlations.

[Figure]

**Figure R1.** Pearson correlation coefficient between the surface (2 m) air temperature of CMA and GLDAS, GLDAS and ERA5, CMA and ERA5, GLDAS and CMFD, CMA and CMFD, ERA5 and CMFD. The black dots denotes the significant ($p < 0.05$) correlations.

2. GLDASv2.0/Noah data is strictly not a climate data record, the authors need to verify the temporal consistency of the used variables to make sure that the trends in the used variables are true reflection of the actual states of Tibetan Plateau and not caused by e.g. the change of the forcing data. A suggestion is compare the relevant variables with the ERA5 data and discuss the uncertainties. Some comparison to in-situ observation should also help to ensure the validity of the conclusions. Examples of such comparisons could be for precipitation (e.g. those from CMA and input to GLDASv2.0/Noah), soil moisture and evaporation from in-situ observation and remote sensing, e.g. https://www.mdpi.com/2072-4292/13/18/3661/htm; https://www.mdpi.com/2072-4292/12/3/509, https://essd.copernicus.org/articles/13/3513/2021, among many others.

**Response 2:** Done. The ERA5 dataset was incorporated to verify the temporal consistency of the used variables to ensure that the shifts and trends in the used variables are true reflection of the actual states of the TP. Moreover, multi-sources of reanalysis, in-situ and remote sensing data were utilized to ensure the reliability of the results. Specifically, the CMFD data were incorporated in precipitation verification. The precipitation of CMA, ERA5, GLDAS and CMFD were inter-compared. The results show that CMA and ERA5 precipitation present high consistency. CMA precipitation has higher consistency with CMFD than ERA5 and GLDAS. Therefore, the precipitation of CMA was adopted in our research (see Figure 4 in the updated manuscript, lines 196-201).

To verify the validity of the soil moisture (SM) data, the in situ SM observations from the International Soil Moisture Network (ISMN; https://ismn.geo.tuwien.ac.at/en/) were used. Figure S2 shows the location of the available 111 ISMN soil moisture stations from three (NGARI, CTP-SMITN, and MAQU) observation networks in this region. The 111 ISMN stations with a data record of 2008-2014 were divided into 0.25° ×0.25° grids. Consequently, 26 measured grids in total were identified with 5 in the arid NAGARI, 12 in the semi-arid and sub-humid CTP-SMITN, and 9 in the humid MAQU. The mean SM value of each grid was obtained by averaging the measurements

of all stations falling within that grid pixel. The validity of GLDAS and ERA5 SM data was verified based on the in situ SM observations from the ISMN. The GLDAS and ERA5 monthly SM were compared with the ISMN measurements over summer periods of May-September from 2008 to 2014 on 25 grids (1 grid without measured data in NAGARI).

Generally, GLDAS shows lower bias and root mean square error (rmse), but higher unbiased rmse (ubrmse) and lower Pearson correlation coefficient (r) than ERA5 (Fig. 2).The ERA5 SM seems more consistent with in situ observations in temporal variations than the GLDAS SM during summer periods. Specifically, Fig. 3 shows that: for MAQU located in humid region, GLDAS SM performs better in bias, but ERA5 is better in r and ubrmse; For NGARI in arid region, ERA5 is better in bias, r, and rmse, but worse in ubrmse than GLDAS; For CTP-SMTMN in semi-arid and sub-humid region, GLDAS shows larger advantage in bias and rmse, but ERA5 has more advantage in r and ubrmse. ERA5 SM seems more advantageous than GLDAS in semi-arid, sub-humid, and humid region without the system bias considered. Both ERA5 and GLDAS SM present considerably higher temporal consistency with in situ measurements in the arid, semi-arid, and sub-humid than humid region (Fig. 3). It may indicate more credibility in SM variation analysis for the arid, semi-arid, and sub-humid regions. (lines 105-109, 180-195 in the updated manuscript)

As the SM variations are largely controlled by the evapotranspiration process, the actual evapotranspiration (ET) of GLDAS and ERA5 was inter-compared (Su et al., 2018). Moreover, the monthly average ET developed by Han et al. (2020, 2021) (hereafter known as Han ET) from 2001 to 2018 was incorporated in the inter-comparison of the GLDAS and ERA5 ET. This dataset is calculated using the surface energy balance system model (SEBS) based on the satellite remote sensing (MODIS) and reanalysis meteorological data (CMFD), and agrees well with the observations of flux towers in the validation of Han et al. (2021). The results (Figure 5) show that the GLDAS and ERA5 ET have high temporal consistency in most TP. The inconsistency mainly in northwest and south fringe of the TP (Fig.5a). The ERA5 ET agrees better with Han ET than that of GLDAS, in particular for the southwest (Fig. 5b,c). The ET

in northwest, north, and southeast seems bearing larger uncertainty than other regions of the TP. (lines 126-129, 201-205 in the updated manuscript)

Based on the above verification and analysis, the ERA5 dataset were incorporated into our analysis of the climate wetting/drying of the TP in the updated manuscript. The performance of ERA5 and GLDAS datasets were inter-compared and the uncertainties were discussed.

[Figure]

**Figure S2.** Location of the in situ soil moisture observations from the International Soil Moisture Network and the climate zone of the TP based on the Aridity index (from http://ref.data.fao.org/map?entryId=221072ae-2090-48a1-be6f5a88f061431a&tab= about).

[Figure]

**Figure 2.** Comparisons of the (a) bias, (b) Pearson correlation coefficient (r), (c) root mean square error (rmse) and (d) unbiased rmse (ubrmse) for GLDAS and ERA5 SM in summer periods of May-September over 2008-2014 based on the soil moisture

measurements from the ISMN.

[Figure]

**Figure 3.** The (a) bias, (b) Pearson correlation coefficient (r), (c) root mean square error (rmse) and (d) unbiased rmse (ubrmse) for GLDAS and ERA5 soil moisture in summer periods of May-September over 2008-2014 based on the ISMN station measurements from the MAQU, NGARI and CTP-SMTMN observation network.

[Figure]

**Figure 5.** Pearson correlation coefficient between the actual evapotranspiration of (a) GLDAS and ERA5, (b) GLDAS and Han ET, and (c) ERA5 and Han ET over summer periods of May-September.

3. Changes in vegetation coverage is closely related to the changes in soil moisture and

temperature, this seems completely neglected by the authors. See e.g. https://link.springer.com/article/10.1007/s10584-009-9787-8.

**Response 3:** Done. The vegetation coverage changes were investigated based on the Normalized Difference Vegetation Index-3rd generation (NDVI) from the Advanced Very High Resolution Radiometer (AVHRR) sensors under the Global Inventory Monitoring and Modeling System (GIMMS). The vegetation coverage presents an increasing trend for the whole TP (Figure R2) and this trend is significant for large regions of north and west TP (Figure 11 in the updated manuscript). The increasing NDVI generally supports the wetting of soil moisture and the overall warming of temperature. The partial correlation analysis between the NDVI and the soil moisture of GLDAS and ERA5 shows that vegetation coverage changes in the east TP tend to be more impacted by temperature, while the changes in the middle and west TP tend to be jointly impacted by temperature and soil moisture, more by soil moisture in some regions, e.g., large regions around 90°E both indicated in GLDAS and ERA5 (Figure 12). The vegetation changes were incorporated in the wetting shift exploration in the revised manuscript as important supports. (lines 298-306 in the updated manuscript)

[Figure]

**Figure R2.** Variations of the yearly average NDVI over summer periods of May–September from 1982 to 2015.

[Figure]

**Figure 11.** Variations of NDVI over summer periods of May–September from 1982 to 2015 for the Tibetan Plateau. The greed/brown △/▽ denotes the significant ($p < 0.05$) increasing/decreasing trends.

[Figure]

**Figure 12.** Spatial distribution of the partial correlations between the NDVI and soil moisture (SM)/surface air temperature (Temp) over summer periods of May–September from 1982 to 2015 for GLDAS and ERA5, respectively. The black dots denote the significant correlations.

4. Figure 4 is rather difficult to comprehend, perhaps a Hovmöller diagram is more effective.

**Response 4:** Done. We tried to use the Hovmöller diagram, but it cannot reflect the spatial distribution and the location changes or the spatiotemporal evolution of SM drought clusters over time. So, Figure 4 in the previous manuscript was revised as follows, which could clearly show the spatiotemporal dynamics of soil moisture drought clusters and the development process of the drought event, as well as the impacts from the precipitation (Prep) and potential evapotranspiration (PET). (see Figure 8 in the updated manuscript)

[Figure]

**Figure 8.** Spatiotemporal dynamic patterns of the most severe SM drought event based on GLDAS over 1961–2014 (the left column), the corresponding monthly standard anomaly maps of Prep (the middle column), and PET (the right column) before (1 month) of the drought duration over the drought cluster affected region. The subgraph in the left bottom marked by "Dynamics from May to Sep", the yellow dots are the centroids of the drought clusters. The black arrow lines show the migration or evolution direction and trajectory of the drought clusters.

5. Trend lines and designated changes in Figures 5, 9 and 11 appear arbitrary, unless the authors can provide more details in trend detection that identifies 1995 as the year of abrupt change. Such a change in Fig. 5 is also not observed nor explained by the monsoon indices, AMO, PDO and ISM. These relationships should be explored more in detail. Does ENSO have any impact on the precipitation and circulation patterns? How about the solar cycles?

**Response 5:** We incorporated more data sources to investigate the abrupt wetting of the

TP. The abrupt wetting was detected in both the drought severity and the average soil moisture, precipitation, and actual evapotranspiration of GLDAS, ERA5 and CMA dataset (Fig. S6). Almost all the aforementioned variables show abrupt changes in the mid to late 1990s at the significance level of $p < 0.05$ based on pettitt diagnosis (pettitt, 1979) (Fig. S7). Supports of the abrupt wetting of the TP can also be drawn from the researches of Zhou et al. (2022), Sun et al. (2020), and Ma and Zhang, 2022. (lines 270-275 in the updated manuscript)

[revised manuscript text omitted]

related to the phase transition of the PDO and the weakening of the ISM and EASM dominated by the ocean oscillation. Additionally, the solar dimming and wind stilling are also contributors of the wetting shift. (lines 353-398 in the updated manuscript)

[Figure]

**Figure S6.** Variations of the total annual soil moisture (SM) drought severity of (a) GLDAS and (b) ERA5, the average SM of (c) GLDAS and (d) ERA5, the yearly average precipitation (Prep) of (e) CMA, (f) GLDAS, and (g) ERA5, and the yearly average actual evapotranspiration (ET) of (h) GLDAS and (i) ERA5 over summer periods (May–September) from 1961 to 2014. The dotted black lines are the mean values over different periods.

[Figure]

**Figure S7.** Pettitt statistics for the total annual soil moisture (SM) drought severity of (a) GLDAS and (b) ERA5, the average SM of (c) GLDAS and (d) ERA5, the yearly average precipitation (Prep) of (e) CMA, (f) GLDAS, and (g) ERA5, and the yearly average actual evapotranspiration (ET) of (h) GLDAS and (i) ERA5 over summer periods (May–September) from 1961 to 2014. The red dotted lines represent the statistics at the significance level of $p = 0.05$.

[Figure]

**Figure 14.** Spatial distribution of the partial correlation coefficients between the precipitation and the (a) AMO, (b) PDO, (c) ENSO (nino 3.4), (d) SRF, (e) EASM and

(f) ISM index on interdecadal scale.

[Figure]

**Figure 15.** Inter decadal variations of the (a) AMO, (b) PDO, (c) ENSO (nino 3.4), (d) SRF, (e) ISM, and (f) EASM index for June–July–August (JJA) over 1961–2014.

[Figure]

**Figure 16.** Differences (Differ) in summer (July) geopotential height (GPH) at 200 hPa between the warm (1995–2014) and cold (1961–1994) phases of the AMO (the later stage minus the former stage). The bold red line represents the border of the TP.

verified based on the in situ SM observations from the ISMN. The GLDAS and ERA5 monthly SM were compared with the ISMN measurements over summer periods of May-September from 2008 to 2014 on 25 grids (1 grid without measured data in NAGARI).

Generally, GLDAS shows lower bias and root mean square error (rmse), but higher unbiased rmse (ubrmse) and lower Pearson correlation coefficient (r) than ERA5 (Fig. 2). ERA5 SM seems more consistent with in situ observations in temporal variations than GLDAS SM during summer periods. Specifically, Fig. 3 shows that: for MAQU located in humid region, GLDAS SM performs better in bias, but ERA5 is better in r and ubrmse; For NGARI in arid region, ERA5 is better in bias, r, and rmse, but worse in ubrmse than GLDAS; For CTP-SMTMN in semi-arid and sub-humid region, GLDAS shows larger advantage in bias and rmse, but ERA5 has more advantage in r and ubrmse. ERA5 SM seems more advantageous than GLDAS in semi-arid, sub-humid, and humid region without the system bias considered. Both ERA5 and GLDAS SM present considerably higher temporal consistency with in situ measurements in arid, semi-arid, and sub-humid than humid region (Fig. 3). It may indicate more credibility in SM variation analysis for the arid, semi-arid, and sub-humid zone. Therefore, to improve the reliability of the results, the ERA5 dataset were incorporated into our investigation of the climate wetting/drying of the TP in the updated manuscript. The performance of ERA5 and GLDAS datasets were inter compared and the uncertainties were discussed. (lines 96-109, 180-195 in the updated manuscript).

[Figure]

**Figure S2.** Location of the in situ soil moisture observations from the International Soil Moisture Network and the climate zone of the TP based on the Aridity index (from http://ref.data.fao.org/map?entryId=221072ae-2090-48a1-be6f5a88f061431a&tab= about).

[Figure]

**Figure 2.** Comparisons of the (a) bias, (b) Pearson correlation coefficient (r), (c) root mean square error (rmse) and (d) unbiased rmse (ubrmse) for GLDAS and ERA5 SM in summer periods of May-September over 2008-2014 based on the soil moisture measurements from the ISMN.

[Figure]

**Figure 3.** The (a) bias, (b) Pearson correlation coefficient (r), (c) root mean square error (rmse) and (d) unbiased rmse (ubrmse) for GLDAS and ERA5 soil moisture in summer periods of May-September over 2008-2014 based on the ISMN station measurements from the MAQU, NGARI and CTP-SMTMN observation network.

2 The language is readable, but still needs to be improved. I found quite some grammar errors. For instance, Line 344-345, "that is" should be "they are".

**Response 2:** Done. The grammar errors were revised carefully and the language was improved in the updated manuscript.

3 The quality of the figures needs to be improved, e.g. Figure 1. And please give full names in figures, for example Dr. Severity and Ave Prep in Figure 5.

**Response 3:** Done. Figure 1 and Figure 5 in the previous manuscript was improved. (Figure 1 and Figure 9 in the updated manuscript).

[Figure]

**Figure 1.** (a) Location of the Tibetan (TP) and the elevation; (b) Monthly average precipitation (Prep), (c) air temperature (Tair) and (d) soil moisture (SM) over the summer (May–September). Prep and Temp are based on the gauging interpolation data provided by the Chinese Meteorological Administration (CMA). SM is from the GLDASv2.0/Noah dataset with a depth of 0-10cm. The black, red, and blue arrows represent the Indian monsoon, the westerlies and the East Asian monsoon, respectively.

[Figure]

Figure 9. Variations of the total annual SM drought severity for (a) GLDAS and (b) ERA5, (c) yearly average precipitation (Prep) of CMA, and potential evapotranspiration (PET) of (d) GLDAS and (e) ERA5 over summer periods (May–September) from 1961 to 2014. The dotted black lines are the average values over periods before and after the mid to late 1990s. The blue and red lines indicate the tendency for periods before and after the mid to late 1990s, respectively.